# SEAgent: Self-Evolving Computer Use Agent with Autonomous Learning from Experience

**Zeyi Sun** [1 2]  **Ziyu Liu** [1 2]  **Yuhang Zang** [2]  **Yuhang Cao** [2]  **Xiaoyi Dong** [2]  **Tong Wu** [3]  **Dahua Lin** [2 4]  **Jiaqi Wang** [5]

## Abstract

Repurposing large vision-language models (LVLMs) as computer use agents (CUAs) has achieved significant breakthroughs, yet these models often struggle with specialized software where human-labeled data is scarce. We propose SEAgent, an agentic self-evolving framework that enables CUAs to autonomously master novel software through experiential learning, exploring unfamiliar applications and progressively tackling auto-generated tasks of increasing complexity. SEAgent combines a World State Model for step-wise trajectory assessment with a Curriculum Generator for task synthesis, and updates the agent's policy via adversarial imitation of failed actions together with Group Relative Policy Optimization (GRPO) on successful ones. We further introduce a specialist-to-generalist training strategy that consolidates individual specialists into a single robust agent, surpassing ensembles of those specialists on their respective software. Across professional applications in OSWorld, Science-Board, and AndroidWorld, SEAgent significantly outperforms the competitive UI-TARS baseline; code and models are publicly available at `https://github.com/SunzeY/SEAgent`.

*"A new generation of agents will acquire superhuman capabilities by learning predominantly from experience."* (Silver & Sutton, 2025)

[1]Shanghai Jiao Tong University, Shanghai, China [2]Shanghai AI Laboratory, Shanghai, China [3]Stanford University, Stanford, CA, USA [4]The Chinese University of Hong Kong, Hong Kong SAR, China [5]Shanghai Innovation Institute, Shanghai, China. Correspondence to: Jiaqi Wang <wangjiaqi@pjlab.org.cn>.

*Proceedings of the 43$^{rd}$ International Conference on Machine Learning*, Seoul, South Korea. PMLR 306, 2026. Copyright 2026 by the author(s).

## 1. Introduction

With the rapid development of large vision-language models (LVLMs) (Touvron et al., 2023; Grattafiori et al., 2024; Bai et al., 2025; Wang et al., 2024; OpenAI, 2023; Anthropic, 2025b; Team et al., 2023), computer use agents (CUAs) (Anthropic, 2024; OpenAI, 2025; Qin et al., 2025; Lin et al., 2024; Wu et al., 2024b) have not only emerged but also demonstrated increasing practical utility. By leveraging the powerful perception and reasoning capabilities of LVLMs, these agents can interpret screenshots as visual inputs and operate computers via keyboard and mouse actions. Despite their promising capabilities, current CUAs (Qi et al., 2024; Putta et al., 2024; Deng et al., 2023; He et al., 2024; Bai et al., 2024; Lu et al., 2024) primarily depend on costly human-curated datasets (Deng et al., 2023; Chen et al., 2024; Wu et al., 2024b; Kapoor et al., 2024; Li et al., 2024), which are typically derived from demonstrations (Lu et al., 2024; Zhang & Zhang, 2023; Gur et al., 2023; Rawles et al., 2023; Zhang et al., 2024a) or video tutorials in the wild (Xu et al., 2024). However, new software continuously emerges and existing software may regularly be updated, often in the absence of annotated human data. It is both necessary and timely to enter an era that emphasizes learning from experience (Silver & Sutton, 2025) in CUA domain. In this paper, we aim to enable CUAs to autonomously explore unfamiliar software environments and evolve into experts without relying on human supervision.

To address this challenge, we propose SEAgent, an agentic self-evolving framework in which Computer Use Agents (CUAs) are exposed to previously unfamiliar software environments and engage in autonomous exploration and experiential learning, as illustrated in Fig. 1. Enabling such self-evolution requires addressing two key challenges: (1) generating executable tasks within unfamiliar software environments, and (2) accurately assessing task success and precisely identifying the step at which failure occurs. To this end, we introduce a **World State Model** for environmental state captioning and step-wise trajectory assessment, together with a **Curriculum Generator** powered by a continuously updated software guidebook memory to generate increasingly diverse and challenging tasks, thereby establishing a curriculum learning paradigm. The agent's policy

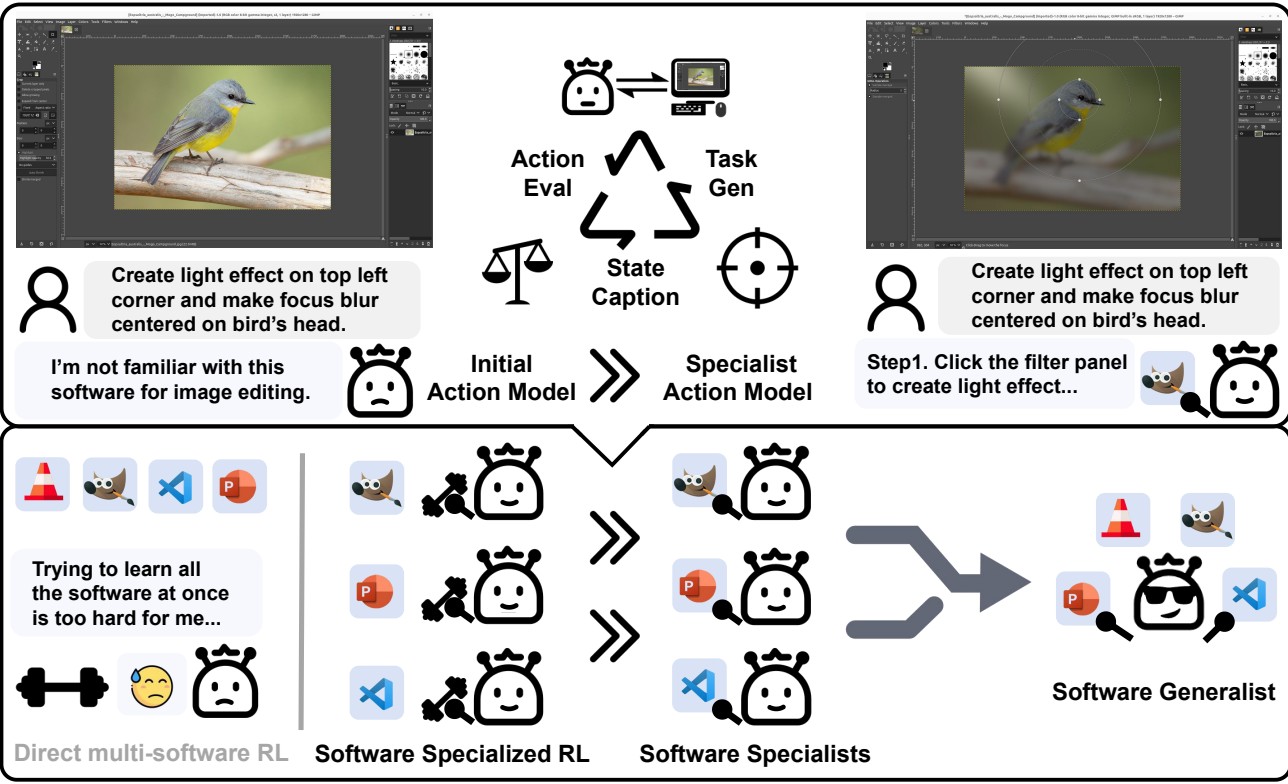

*Figure 1.* **SEAgent enables computer use agents self-evolving in novel environments** by autonomously exploring and learning from their own experiences without human intervention. The specialist-to-generalist training strategy further enhances the development of a strong generalist agent.

is optimized through experiential learning from both failures and successes, combining adversarial imitation of failure actions and Group Relative Policy Optimization (GRPO) on successful ones.

Given the critical role of reward accuracy, we conduct extensive evaluations and observe that existing reward models of computer use tasks fall short in terms of judgment precision and reward density. Leveraging the enhanced long-context processing capabilities of advanced LVLMs, we input the agent's full trajectory of states into the reward model and fine-tune a reward model, World State Model, using Qwen2.5-VL (Bai et al., 2025), substantially narrowing the gap with commercial models such as GPT-4o (OpenAI, 2023) with +7.5% improvement in precision compared to baseline model in evaluating CUAs' trajectories on AgentRewardBench (Lù et al., 2025), enable World State Model to provide high quality step level reward signals in self-evolving agentic system.

Moreover, SEAgent enables agents to evolve into either single-software specialists or multi-software generalists. To overcome the limitation that directly training a generalist underperforms compared to specialists, inspired by (Zhang et al., 2024c), we introduce a novel specialist-to-generalist training strategy, which even surpasses the performance of individual specialists on their respective software applica-

tions. We perform extensive experiments of SEAgent built on UI-TARS (Qin et al., 2025) and evaluated on five professional software applications from OSWorld (Xie et al., 2024). SEAgent with the specialist-to-generalist strategy significantly improves the UI-TARS (Qin et al., 2025). Furthermore, SEAgent with the specialist-to-generalist strategy outperforms both specialist RL and generalist RL by a substantial margin, demonstrating the effectiveness of the specialist-to-generalist paradigm. We also validate this strategy on UI-TARS-1.5 on ScienceBoard (Sun et al., 2025) on out of domain scientific softwares. In general, SEAgent offers a promising approach for developing more powerful and versatile computer-use agents without human involvement.

**Conflict of Interest Disclosure.** The authors have no financial conflict of interest to disclose. SEAgent is built on top of the open-source UI-TARS and Qwen-VL series and is evaluated on third-party benchmarks (OSWorld, AndroidWorld, ScienceBoard, AgentRewardBench), none of which are developed by, or commercially associated with, the institutions that employ the authors.

## 2. Related Work

**Agent for Computer Use.** Advancements in LLMs and LVLMs (Touvron et al., 2023; Grattafiori et al., 2024; Liu

et al., 2023a; Bai et al., 2025; Wang et al., 2024) have catalyzed research into computer use agents (CUAs) (Hu et al., 2024; Hong et al., 2024; Cheng et al., 2024; Nguyen et al., 2024; Lin et al., 2024; Wang et al., 2025; Ye et al., 2025) that process structured text (Qi et al., 2024) or screenshots. While performing well in-domain (Lu et al., 2024; Zheng et al., 2024; Liu et al., 2024; Li et al., 2025; Cheng et al., 2024), CUAs still struggle in complex simulations (Xie et al., 2024; Rawles et al., 2024; Koh et al., 2024; Zhou et al., 2023). To improve grounding and decision-making, prior works explored expert decomposition (Gou et al., 2024; Wan et al., 2024) and agent collaboration (Agashe et al., 2024; 2025; Liu et al., 2023b; Zhang et al., 2025; Wang et al., 2025) via prompt engineering (Yan et al., 2023; He et al., 2024; Zhang et al., 2024b; Wang et al., 2023; Wu et al., 2024a), while recent foundational GUI agents (Ye et al., 2025; Wang et al., 2025) consolidate perception, planning, and action into a single model trained on large-scale curated GUI corpora. However, these training-free or supervised methods still rely heavily on human-curated data and offer limited task-specific adaptation in unfamiliar software. Our work focuses on reinforcement fine-tuning, enabling CUAs to achieve autonomous evolution on novel software without human annotations.

**Reinforcement Learning for LLM/LVLMs.** Posttraining has shifted from data-intensive SFT (Liu et al., 2023a; Wei et al., 2022) toward Reinforcement Learning (RL) (Touvron et al., 2023; Grattafiori et al., 2024; Liu et al., 2023a; Bai et al., 2025; Wang et al., 2024). Despite the success of single-turn RLHF (Ouyang et al., 2022; Ziegler et al., 2019; Rafailov et al., 2023; Luo et al., 2025) and GRPO (Guo et al., 2025; Shao et al., 2024), multi-step agentic tasks remain difficult due to sparse feedback. Existing RL methods for agents (Bai et al., 2024; Qi et al., 2024; Zhou et al., 2024; Zhai et al., 2024; Carta et al., 2023; Chen et al., 2025; Lu et al., 2025) typically rely on dedicated critic models for advantage estimation, end-to-end policy optimization with experience replay (Lu et al., 2025), or DPO on interaction data (Putta et al., 2024; Qin et al., 2025). In contrast, we evaluate diverse reward modeling strategies and demonstrate that step-wise reward analysis offers denser and more accurate credit assignment for long-horizon CUAs than trajectory-level critic-based estimation (Bai et al., 2024; Qi et al., 2024; Lu et al., 2025), while remaining complementary to trajectory-level RL when the evaluation budget is sufficiently fault-tolerant.

## 3. Methods

**Problem Formulation.** The objective of SEAgent is to establish a training pipeline enabling the Computer Use Agent (CUA) to autonomously explore its environment (Sec. 3.1) and progressively self-evolve on novel software applica-

tions via reinforcement learning from experience (Sec. 3.2). Specifically, as illustrated in left part of Fig. 2, the SEAgent pipeline comprises three primary components: an Actor Model $\pi$ performing exploratory actions to accomplish these tasks, and a World State Model $\mathcal{M}_{state}$ describing the current environment state and evaluating the success or failure of executed actions, and a Curriculum Generator $\mathcal{M}_{task}$ that continuously proposes more diverse and challenging exploration tasks:

**(1) Actor Model $\pi$:** The policy $\pi(a|s_t, I)$ defines the probability of taking action $a$ at time step $t$, conditioned on the current state $s_t$ and the overall task instruction $I$.

**(2) World State Model $\mathcal{M}_{state}$:** This component is a fine-tuned Large Vision-Language Model (LVLM) responsible for providing detailed descriptions of environment states. It also evaluates each step of the trajectory executed by the Actor Model $\pi$, producing trajectory judgement $\mathcal{J}$ which indicates whether the task has been successfully completed. Joint training with state change captioning $\mathcal{C}$ of the software GUI has been shown to enhance judgment accuracy, as shown in Table 1.

**(3) Curriculum Generator $\mathcal{M}_{task}$:** This component utilizes a powerful Large Language Model (LLM) to automatically generate novel exploration tasks. It also maintains and updates a software guidebook $U$ based on the state change captioning $\mathcal{C}$ and the trajectory judgement $\mathcal{J}$ provided by $\mathcal{M}_{state}$ during interactions. The gradually enriched guidebook $U$ enables $\mathcal{M}_{task}$ to progressively generate increasingly diverse and challenging tasks in a curriculum learning fashion.

SEAgent can be applied to enable the self-evolution of a computer-use agent, either as a specialist for a single software or as a generalist across multiple software. However, we observe that direct training for generalist agents is suboptimal. We introduce a specialist-to-generalist training strategy, which achieves improved overall performance than training multiple generalist agents detailed in Sec. 3.3.

### 3.1. Autonomous Exploration with Self-evolving Curriculum

Autonomous exploration is essential for enabling the Computer Use Agent (CUA) to develop proficiency in novel software applications that are previously unseen or poorly understood. This process involves addressing two key challenges: (1) generating executable tasks within unfamiliar software environments, and (2) evaluating task completion success and pinpointing the specific step at which failure occurs. To tackle these challenges, we introduce two novel components: the World State Model $\mathcal{M}_{state}$ and the Curriculum Generator $\mathcal{M}_{task}$. These components jointly support a **self-evolving curriculum paradigm**, which facilitates

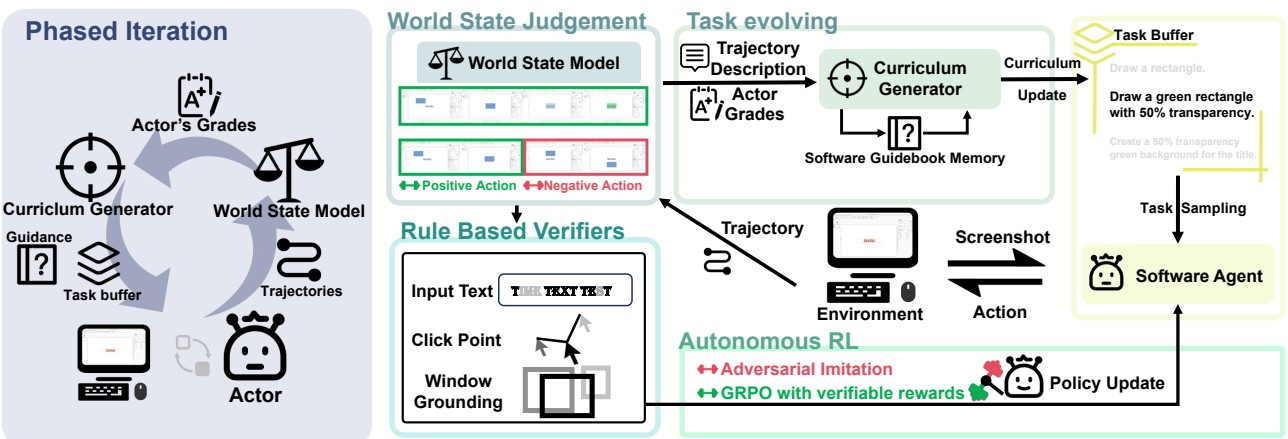

Figure 2. **SEAgent autonomous exploration and experiential learning pipeline.** Guided by tasks generated by the Curriculum Generator, the Actor Model is updated according to step-level rewards from the World State Model through verifiable reward functions tailored for different action types.

the autonomous generation of increasingly diverse and challenging tasks.

The **self-evolving curriculum paradigm** pipeline is structured into $P$ sequential phases. Before the first phase, a set of initial tasks targeting basic GUI operations is generated (details provided in Sup. C.1). In each phase, these tasks are executed and step-wise evaluated. The resulting judgments and descriptions of the exploration trajectories are fed into the Curriculum Generator $\mathcal{M}_{\text{task}}$, which updates a self-maintained software guidebook $U$. Leveraging this updated guidebook and the current capabilities of the CUA, the generator then produces more diverse and challenging tasks for subsequent phases. The following outlines each step of the process in detail:

**(1) Task initiation:** The initial state of the unfamiliar software is provided, typically in the form of screenshots of its basic GUI interface. The World State Model $\mathcal{M}_{\text{state}}$ performs dense captioning of the GUI elements, including button detection and OCR-based recognition. These detailed captions are passed to the Curriculum Generator $\mathcal{M}_{\text{task}}$, which generates an initial set of task instructions $\mathcal{I}_0 = \{I_0^{(1)}, I_0^{(2)}, \cdots\}$ along with an initial software guidebook $U_0$ for the software.

**(2) World state judgment:** In the $p$-th phase of *Auto Exploration*, the Actor Model $\pi_p$ executes tasks based on the instructions in $\mathcal{I}_p$. Each execution is evaluated by the World State Model $\mathcal{M}_{\text{state}}$, which provides feedback $\mathcal{J}_p = \{J_p^{(1)}, J_p^{(2)}, \cdots\}$ for each step within the operation trajectory. In addition, it generates a detailed description of GUI state changes based on captured screenshots, denoted as $\mathcal{C}_p$.

**(3) Task self-evolving:** As shown in right part of Fig. 2, based on the outcomes $\mathcal{J}_p$ and $\mathcal{C}_p$, the Curriculum Generator $\mathcal{M}_{\text{task}}$ produces a more challenging task set $\mathcal{I}_{p+1}$ and

expands the agent's knowledge boundary by updating the software guidebook to $U_{p+1}$. The detailed prompting process is illustrated in Fig. 9. This iterative update can be formalized as:

$$U_{p+1}, \mathcal{I}_{p+1} = \mathcal{M}_{\text{task}}(U_p, \mathcal{I}_p, \mathcal{J}_p, \mathcal{C}_p) \quad (1)$$

Here, $U_{p+1}$ serves as a more comprehensive software guidebook memory, while $\mathcal{I}_{p+1}$ represents a more challenging task set tailored to the current capabilities of the Actor Model $\pi_p$. Examples of $\mathcal{I}_p$ are provided in Fig. 4, where the Actor Model $\pi$ demonstrates curriculum learning by handling increasingly complex tasks across different phases $p$. Illustrations of $U_p$ across various software applications are provided in Sup. K. Comparison with previous methods (Murty et al., 2025; 2024; Sun et al., 2024) on task generation are detailed in Sup.C.2

**(4) Autonomous RL Training:** Through iterative reinforcement learning, the Actor Model $\pi_p$ is updated based on its execution of the instruction set $\mathcal{I}_p$, guided by evaluation feedback $\mathcal{J}_p$ and a set of action-specific verifiable functions $\mathcal{R}_{\text{verifer}}$. The resulting policy $\pi_{p+1}$ is then used as the actor in the subsequent phase. Further details are provided in Sec. 3.2.

### 3.2. Reinforcement Learning from Experience

The World State Model $\mathcal{M}_{state}$ provides step-level reward signals for reinforcement learning. Unlike previous reward models for CUA (Qi et al., 2024; Bai et al., 2024; Putta et al., 2024; Pan et al., 2024; Lù et al., 2025), our $\mathcal{M}_{state}$ model takes the entire trajectory of states and actions, $\mathcal{H} = \{(s_0, a_0), (s_1, a_1), \ldots\}$, as input. It classifies each action $a$ as either $a_F$ or $a_T$, where $a_F$ indicates an incorrect action leading to failure or redundant loops, and $a_T$ represents a correct action that contributes to successful progression without redundancy. The curated prompt used

for judgment is depicted in Fig. 8 with input/output format detailed in Sec. A.1. For historical states that result in $a_T$, we encourage CUA to reinforce these actions through verifiable rewards defined by a set of functions $\mathcal{R}_{\text{verifer}} = \{r_{dist}\}$. Conversely, for states leading to $a_F$, we penalize them using negative KL divergence with adversarial imitation.

**Adversarial Imitation for Failure Action Punishment.** To explicitly encourage the policy to diverge from failure-inducing behaviors, we employ a contrastive log-ratio loss based on a reference failure action $a_F$. This objective encourages the policy to sample actions $a$ that minimize alignment with the failure action $a_F$:

$$\mathcal{L}_{\text{AI}}(\pi_\theta) = \mathbb{E}_\nu \left[ -\log \frac{\pi_\theta(a \mid s, I)}{\pi_{\text{ref}}(a_F \mid s, I)} \right] \tag{2}$$

**Verifiable Rewards for Correct Action Encouragement.** To more effectively guide the policy toward correct actions $a_T$, we adopt Reinforcement Learning with Verifiable Rewards (RLVR) (Guo et al., 2025; Shao et al., 2024), which has recently shown success in enhancing language models on tasks with objectively verifiable answers, such as mathematics (Shao et al., 2024), and more recently, counting and grounding in the vision-language domain (Liu et al., 2025b; Shen et al., 2025; Meng et al., 2025). After labeling the correct step $(s, a_T)$ using the World State Model, we apply Group Relative Policy Optimization (GRPO), computing the relative advantage of each response based on its reward:

$$A^{(i)} = \frac{r^{(i)} - \text{mean}(\{r^{(j)}\}_{j=1}^G)}{\text{std}(\{r^{(j)}\}_{j=1}^G)}, \quad i = 1, \cdots, G. \tag{3}$$

As we design distinct reward signals for different action types, we define the reward function between a predicted action $a$ and the ground-truth action $a_T$ as:

$$r^{(i)} = r(a^{(i)}, a_T) = \mathbb{I}\left(\text{type}(a^{(i)}) = \text{type}(a_T)\right) \\ + r_{\text{dist}}(a^{(i)}, a_T), \tag{4}$$

where $\mathbb{I}(\cdot)$ is the indicator function that returns 1 if the predicted action and ground-truth action are of the same type, and 0 otherwise. The distance-based reward term $r_{\text{dist}}(a^{(i)}, a_T)$ is defined according to the specific action type: for `click` actions, it is computed based on the normalized L1 distance between the clicked coordinates; for `drag` and `select` actions, it is computed using the Intersection over Union (IoU) between the predicted and ground-truth bounding boxes; and for `type` actions, it is determined by the character-level BLEU score between the predicted and ground-truth text. All $r_{\text{dist}}$ rewards are normalized to the range $[0, 1]$ to ensure consistency across different action types. A comprehensive list of $r_{\text{dist}}(a^{(i)}, a_T)$ definitions for

various action types is provided in Tab. 10. The final loss of GRPO is directly adopted from (Shao et al., 2024):

$$\mathcal{L}_{\text{GRPO}}(\pi_\theta) = -\mathbb{E}_{(s,I)\sim\mathcal{D}, \{a^{(i)}\}_{i=1}^G \sim \pi_{\text{ref}}(\cdot|s,I)} \tag{5}$$

$$\left[ \frac{1}{G} \sum_{i=1}^G \frac{1}{|a^{(i)}|} \sum_{t=1}^{|a^{(i)}|} \left\{ \min\left(r_t^{(i)}(\theta)A^{(i)}, \text{clip}(r_t^{(i)}(\theta), \right. \right. \tag{6}$$

$$\left. \left. 1 - \epsilon, 1 + \epsilon)A^{(i)}\right) - \beta\, D_{\text{KL}}^{(i,t)}(\pi_\theta \| \pi_{\text{ref}}) \right\} \right],$$

where $r^{i,t}(\theta) = \frac{\pi_\theta(a^{(i)}|s,I)}{\pi_{\theta_{\text{ref}}}(a^{(i)}|s,I)}$ and $D_{\text{KL}}^{i,t}(\pi_\theta, \pi_{\text{ref}})$

$$= \frac{\pi_{\text{ref}}(a^{(i)}|s,I)}{\pi_\theta(a^{(i)}|s,I)} - 1 - \log\frac{\pi_{\text{ref}}(a^{(i)}|s,I)}{\pi_\theta(a^{(i)}|s,I)}.$$

Similar to (Shao et al., 2024; Guo et al., 2025), advantage $A$ is weighted on the whole reasoning token logits to encourage free form thinking for performing action and planning.

The final training loss is defined as a weighted combination of positive and negative action samples, i.e., correct actions $a_T$ and incorrect actions $a_F$: $\mathcal{L}(\pi(\theta)) = \mathcal{L}_{\text{GRPO}} + \gamma \mathcal{L}_{\text{AI}}$. We set $\gamma = 0.2$ during training, and the rationale for this choice is discussed in the ablation study presented in Sup. G.

This strategy is shown to be more effective in Sec. 4.2 compared to Generalized Advantage Estimation (GAE) (Schulman et al., 2015)-based RL methods (Qi et al., 2024; Bai et al., 2024), as the more powerful reward model $\mathcal{M}_{state}$ provides accurate step-level reward signals by leveraging the entire episode trajectory $\mathcal{H}$ from a global perspective.

**Why step-wise rather than trajectory-wise.** Computer-use training in unfamiliar software is long-horizon and extremely sparse-reward, so trajectory-level credit assignment is noisy and sample-inefficient. Our step-wise design replaces the binary $0/1$ trajectory signal with a continuous per-step reward in $[0, 1]$ computed jointly from the action type and parameters (Eq. 4), allowing the policy to be optimized at the same granularity as the reward. The WSM evaluates whether the observed state transition reflects task progress rather than enforcing exact action matching, so the step-wise signal trades a small amount of action diversity for substantially lower-variance credit assignment. Sec. 4.2 verifies this design against trajectory-level variants under the protocol of Lu et al. (2025): step-wise RL is more effective under the standard step-limited setting, while trajectory-level RL becomes preferable only when the interaction horizon is large enough to leverage error recovery.

### 3.3. From Specialist to Generalist.

Achieving a generalist agent capable of operating across multiple software platforms is an ambitious and valuable goal. We first attempted to train such a generalist directly using the proposed SEAgent framework across all software environments. However, this approach led to suboptimal performance compared to specialized agents, as the actor struggled to learn effectively in the multi-software environment.

We thus introduce a specialist to generalist strategy, as illustrated in Fig. 1. Specifically, we first train software-specialized agents via SEAgent on individual environments, allowing each to master a specific application. These specialists are then distilled into a single generalist model through supervised fine-tuning (SFT) on synthesized successful trajectories. Finally, the generalist is refined via SEAgent on multiple software. This generalist, now equipped with better reasoning, planning abilities, and software-specific commonsense, achieves significantly improved performance, outperforming both the SEAgent via direct general RL and the performance combination of multi-specialists as in Table 2.

## 4. Experiments

### 4.1. Evaluation of Reward Model for GUI agent.

Providing GUI agents with reliable reward signals is crucial for enabling self-evolution. Building on AgentReward-Bench (Lù et al., 2025), which focuses on web tasks, we extend the evaluation to a broader set of PC software environments. Specifically, we evaluate on all 339 feasible tasks from OSWorld (Xie et al., 2024). Trajectories are sampled from UI-TARS (Qin et al., 2025) and Gemini-2.5-Pro (Google DeepMind, 2025b), and a rule-based evaluation is used as ground-truth supervision to compute a confusion matrix for each reward model's predictions.

The judging strategy in AgentRewardBench (Lù et al., 2025) is limited, as it relies solely on the final state and action history. It is more natural and reliable for a judge model to consider the entire trajectory when assessing task success as also adopted in EPO (Liu et al., 2025a). For example, a final state message like "Your flight ticket has been successfully booked" does not confirm whether the correct date and time were selected, which can lead to compromised judgment accuracy.

However, we observe that current open-sourced LVLMs perform poorly under this more holistic evaluation strategy. As shown in Fig. 3, feeding additional historical screenshots into Qwen2.5-VL (Bai et al., 2025) significantly degrades its Average Precision (AP), diverging notably from GPT-4o (Hurst et al., 2024) on the same curated prompt. We

attribute this performance drop to the insufficient pretraining of Qwen2.5-VL on long sequences of high-resolution screenshots, which pushes it toward the limits of its 32K context length.

To address this, we propose World State Model, a distilled model based on Qwen2.5-VL-7B. The training process for World State Model uses a dataset of 0.86K GPT-4o (Hurst et al., 2024) generated evaluations on trajectories with dense GUI change descriptions, exclusively from the Chrome browser within the OSWorld (Xie et al., 2024) environment. Alongside the primary judgment task, we also find it effective to co-training the model with change description (CD) task for describing the difference of the screenshot before and after an action. Training data and settings are detailed in Sup. A. Despite being trained solely on Chrome data, World State Model exhibits strong generalization to other professional software in OSWorld and to the external AgentRewardBench (Lù et al., 2025) benchmark. This demonstrates that the model learns transferable judgment patterns rather than overfitting to a single application.

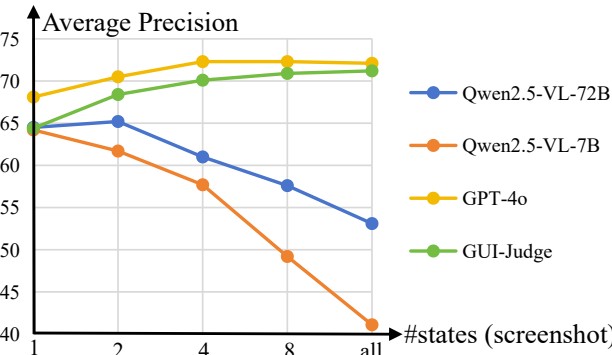

*Figure 3.* **The Average Precision on AgentRewardBench (Lù et al., 2025)**, where GUI-Judge exhibits an improvement in AP as the number of input middle states increases, showing a similar trend to that of the closed sourced GPT-4o.

As evaluated in Tab. 1 and further analyzed in Fig. 3, World State Model achieves state-of-the-art performance among open-sourced judges and significantly narrows the gap with proprietary alternatives such as GPT-4o (Hurst et al., 2024) and Gemini-3 Pro (Google DeepMind, 2025a). To rule out that the choice of teacher model dominates the result, we additionally distill World State Model from Gemini-3 Pro using the same 860 trajectories. The two distilled variants exhibit different but complementary trade-offs: Gemini-3-Pro-distilled World State Model is consistently slightly better in precision (e.g., 74.2 vs. 73.9 on OSWorld-Full and 71.5 vs. 69.3 on Prof/Office), whereas GPT-4o-distilled World State Model is stronger in NPV. We attribute this to the underlying teachers themselves: Gemini-3 Pro is a stricter judge (some trajectories that it labels as failures still pass the relatively permissive OSWorld verifier, lowering NPV), while GPT-4o is more permissive. The bottleneck in distillation is closer to the student capacity (Qwen2.5-VL-7B) than to teacher quality, which is why the gain from

*Table 1.* **Precision and Negative Predictive Value (NPV)** on AgentRewardBench (Lù et al., 2025) and OSWorld (Xie et al., 2024) with last screenshot only (LS) or entire process screenshots (ES) as input. World State Model closes the gap with proprietary judges (GPT-4o, Gemini-3 Pro). Co-training with screenshot change description (CD) further improves judgment precision. The two distilled World State Model variants (from GPT-4o and from Gemini-3 Pro) are complementary: GPT-4o-distilled is stronger in NPV, while Gemini-3-Pro-distilled is stronger in precision.

| Model | Input | AgentRewardBench (Lù et al., 2025) | | OS-World-Full | | Prof/Office | |
| | | Precision | NPV | Precision | NPV | Precision | NPV |
|---|---|---|---|---|---|---|---|
| GPT-4o (Hurst et al., 2024) | LS | 68.1 | 92.3 | 46.3 | 88.2 | 40.5 | 81.0 |
| | ES | 72.1 | 92.2 | 74.6 | 95.2 | 70.4 | 85.3 |
| Gemini-3 Pro (Google DeepMind, 2025a) | ES | 83.1 | 91.5 | 79.9 | 90.2 | 77.3 | 83.5 |
| Qwen2.5-VL-72B (Bai et al., 2025) | LS | 64.5 | 94.2 | 41.5 | 86.9 | 31.7 | 78.7 |
| | ES | 26.2 | 83.0 | 26.8 | 83.0 | 25.6 | 76.6 |
| Qwen2.5-VL-7B (Bai et al., 2025) | LS | 64.1 | 90.3 | 37.3 | 85.2 | 31.8 | 79.0 |
| | ES | 25.4 | 83.8 | 20.0 | 81.7 | 23.5 | 76.0 |
| World State Model (w/o CD) | LS | 66.8 | 90.6 | 52.5 | 85.8 | 57.1 | 81.1 |
| World State Model (w/ CD) | LS | 65.2 | 79.1 | 49.3 | 82.0 | 55.8 | 77.2 |
| World State Model (w/o CD) | ES | 69.1 | 88.5 | 71.1 | 88.4 | 65.0 | 81.1 |
| World State Model (w/ CD, GPT-4o distilled) | ES | 71.6 | **91.2** | 73.9 | **90.5** | 69.3 | **82.0** |
| World State Model (w/ CD, Gemini-3 Pro distilled) | ES | **72.1** | 89.8 | **74.2** | 89.4 | **71.5** | 82.3 |

a stronger teacher is small after distillation. Both distilled variants generalize beyond the Chrome-only training source to OSWorld professional/office software, validating that the WSM learns transferable judgment patterns. We adopt the GPT-4o-distilled variant in the main pipeline because higher NPV is more important when the AI loss (Sec. 3.2) reuses failure-labeled steps: an overly strict judge would over-penalize weak agents by labeling too many recoverable trajectories as failures. Finally, the resulting open-source World State Model provides reliable step-level reward signals (Fig. 5), allowing our agentic system to evolve without expensive API calls to proprietary models.

## 4.2. Self evolution of GUI Agents

**Models Before Self-Evolution.** Our self-evolving system is initialized with three locally deployed models: UI-TARS-7B-DPO (Qin et al., 2025) as the Actor Model, World State Model as the step-level reward model, and Qwen2.5-72B (Yang et al., 2024) as the Curriculum Generator. As shown in Tab. 2, the initial Actor Agent achieves an average success rate of 21.5% across five professional software applications from OSWorld.

**Evolution Process Details.** The evolution process begins with the Curriculum Generator producing an initial instruction set ($\mathcal{I}_0$), averaging 150.2 instructions. The Actor Model executes these tasks, and the resulting trajectories are evaluated by World State Model and parsed into an average of 1361.5 multi-turn conversation pairs (detailed statistics are in Sup.I). We then perform reinforcement fine-tuning (RFT) for 1k iterations on 8 NVIDIA A100 80GB GPUs, with a batch size of 16 and a learning rate of $2 \times 10^{-5}$ scheduled via cosine decay. This process is repeated for three phases.

**Specialist Evaluation.** For a fair comparison with previous methods (Bai et al., 2024; Qi et al., 2024), we train

specialist agents for five different software applications. We adapt their strategies by initializing a separate critic model from UI-TARS-7B with randomly initialized MLP layers to regress value predictions using Generalized Advantage Estimation (GAE) (Schulman et al., 2015). As shown in Tab. 2 and Fig. 4, SEAgent, achieves superior performance. We attribute this to World State Model providing fine-grained, step-level rewards from the full history, which is more effective than relying on a separate critic to estimate advantages from sparse, final success/failure signals. Experimental results on mobile use agent are supplied in Sec. E. We also provide comparison with previous task generation methods (Murty et al., 2025; Qi et al., 2024) on task generation are detailed in Sup.C.2.

As shown in Fig. 4 and Tab. 2, we train separate actor agents for five different software applications. Our approach, denoted as SEAgent (Specialist), achieves strong performance compared to previous reinforcement learning methods such as DigiRL (Bai et al., 2024) and WebRL (Qi et al., 2024). We attribute this improvement to the use of World State Model, which provides fine-grained, step-level reward signals derived from a comprehensive understanding of the full history of states and actions. This contrasts with previous approaches that rely on separate critic models—typically initialized from the actor itself—to estimate advantages from sparse, final success/failure signals. Furthermore, the curriculum of task instructions generated by the Curriculum Generator, as illustrated in Fig. 4, validates the effectiveness of our autonomous learning framework. These tasks progress from simple to complex based on the actor's evolving capabilities, enabling it to gradually specialize in each target software environment. Based on the observed evolution curves, we set the number of training phases to three, as performance gains saturate beyond that point.

**From Specialist to Generalist.** After training five strong software specialists, we pursue generalization. We collect

*Table 2.* **Success Rate (SR) on OSWorld (Xie et al., 2024)**. SEAgent demonstrates strong performance after reinforcement learning. In addition to evolving on separate software, a new General Model achieves better performance after another iteration of SEAgent. *Indicates specialist agents trained separately for each software with ensembled results. All results are averaged over five runs.

| Model | VScode | GIMP | Impress | VLC | Writer | Overall |
|---|---|---|---|---|---|---|
| Human Performance | 73.9 | 73.1 | 80.9 | 70.6 | 73.9 | 74.5 |
| GPT-4o (Hurst et al., 2024) | 4.35 | 3.85 | 6.77 | 16.1 | 4.35 | 7.08 |
| GPT-4V (OpenAI, 2023) | 0.00 | 7.69 | 2.52 | 18.3 | 4.35 | 6.59 |
| Gemini-Pro-1.5 (Team et al., 2023) | 0.00 | 11.5 | 13.2 | 6.53 | 8.71 | 7.99 |
| Claude3.7 Sonnet (Anthropic, 2025a) | 18.8 | 24.4 | 10.6 | 27.5 | 17.4 | 19.7 |
| Gemini-Pro-2.5 (Google DeepMind, 2025b) | 21.7 | 26.9 | 9.92 | 25.5 | 24.6 | 21.7 |
| UI-TARS-7B-DPO (Lu et al., 2024) | 30.4 | 34.6 | 17.0 | 11.8 | 13.6 | 21.5 |
| UI-TARS-72B-DPO (Lu et al., 2024) | 39.1 | 53.8 | 23.4 | 15.3 | 26.1 | 31.5 |
| DigiRL (Bai et al., 2024) (Specialized RL)* | 43.7 | 45.4 | 19.6 | 25.0 | 19.1 | 30.6 |
| WebRL (Qi et al., 2024) (Specialized RL)* | 36.5 | 37.7 | 20.4 | 29.4 | 21.7 | 29.1 |
| SEAgent (Specialized RL)* | 46.1 | 50.0 | 21.3 | 31.8 | 33.0 | 36.4 |
| DigiRL (Bai et al., 2024) (General RL) | 38.3 | 46.2 | 19.1 | 25.9 | 19.1 | 29.7 |
| WebRL (Qi et al., 2024) (General RL) | 35.6 | 33.1 | 18.7 | 27.0 | 15.7 | 26.0 |
| SEAgent (General RL) | 40.8 | 42.3 | 21.7 | 28.2 | 30.4 | 32.6 |
| SEAgent (General SFT) | 36.5 | 41.5 | 25.5 | 30.6 | 32.2 | 33.3 |
| SEAgent (Specialist-to-Generalist) | **47.8** | **50.8** | **29.8** | **35.3** | **36.5** | **40.0** |

*Table 3.* **Success Rate (SR)** on OSWorld (Xie et al., 2024) and ScienceBoard (Sun et al., 2025) with UI-TARS-1.5 (Qin et al., 2025).

| Benchmark | OSWorld (Xie et al., 2024) | | | | | ScienceBoard (Sun et al., 2025) | | | |
|---|---|---|---|---|---|---|---|---|---|
| *Software* | Impress | Writer | GIMP | VScode | VLC | ChamerX | GrassGIS | KAlgebra | Celestia |
| UI-TARS-1.5-7B-DPO | 29.8 | 39.1 | 51.5 | 60.9 | 23.5 | 12.4 | 0.0 | 11.6 | 4.9 |
| UI-TARS-1.5-7B-DPO + SEAgent | 31.9 | 43.5 | 56.9 | 60.9 | 35.3 | 31.0 | 20.6 | 29.0 | 15.2 |

task instructions from each specialist's training and use them to generate 3.5K successful trajectories. These trajectories, along with their reasoning traces, are distilled into a new base model (UI-TARS-7B) via supervised fine-tuning (SFT). This distilled model is then further optimized through RL across all five software environments. As shown in Tab. 2, the resulting generalist model surpasses the performance of the individual specialist ensemble. To ensure a fair comparison, we establish a "General RL" baseline, which involves training an agent directly across five software applications. We then provide this baseline with an additional round of reinforcement learning, configuring it with a step count and data volume identical to those of the Specialist-to-Generalist setting. This resource-matched comparison proves that the generalization-through-specialization approach (Fig. 1) leads to superior performance.

**Cross-Environment Generalization.** Our work targets out-of-domain (OOD) software where human-labeled data is not available. We therefore apply SEAgent to UI-TARS-1.5 on two benchmarks (Tab. 3). On OSWorld (Xie et al., 2024), an in-domain environment for UI-TARS-1.5, we observe moderate gains, while on ScienceBoard (Sun et al., 2025)—a suite of scientific applications that are truly novel to UI-TARS-1.5—SEAgent delivers substantial improvements (e.g., $0.0 \rightarrow 20.6$ on GrassGIS and $4.9 \rightarrow 15.2$ on Celestia), validating our core claim that the framework is most impactful for self-evolution on truly OOD software. As an additional sanity check on transfer, we directly evaluate

*Table 4.* Ablation across three OSWorld (Xie et al., 2024) environments. Using World State Model yields consistent gains over Qwen2.5VL-72B (Bai et al., 2025) as the reward model. We further ablate supervised fine-tuning (behavior cloning, BC), GRPO, and Adversarial Imitation (AI). The first row uses the base UI-TARS-7B-DPO without any reward model. The full system performs best on all three environments.

| Qwen-72B | WSM | BC | GRPO | AI | VScode | GIMP | VLC |
|---|---|---|---|---|---|---|---|
| | | | | | 30.4 | 34.6 | 11.8 |
| ✓ | | ✓ | | | 26.1 | 32.3 | 13.7 |
| ✓ | | | ✓ | | 28.3 | 33.8 | 11.8 |
| | | ✓ | ✓ | | 34.8 | 37.7 | 15.7 |
| | ✓ | ✓ | | ✓ | 39.1 | 38.5 | 19.6 |
| | ✓ | | ✓ | | 43.5 | 42.3 | 25.5 |
| | ✓ | | ✓ | ✓ | **46.1** | **50.0** | **31.8** |

the OSWorld-trained UI-TARS-7B-DPO agent on Science-Board and AndroidWorld (Rawles et al., 2024) *without any further finetuning*: the base model achieves $3.4\%/33.0\%$ on the two benchmarks, while SEAgent improves it to $11.3\%/35.5\%$. The same WSM, distilled only from 860 Chrome trajectories, also transfers to AndroidWorld, with judgment precision rising from $41.9\%$ to $58.4\%$ after WSM training. We exclude the Lean and TeX applications from ScienceBoard as their text- and code-based interfaces are unsuitable for a GUI-centric agent.

**Ablation Studies.** We ablate the contribution of each component of SEAgent in Tab. 4 on three OSWorld (Xie et al., 2024) environments (VSCode, GIMP, VLC). Two consistent trends emerge across all three software. First, replacing Qwen2.5-VL-72B with our World State Model as the reward model substantially improves downstream training,

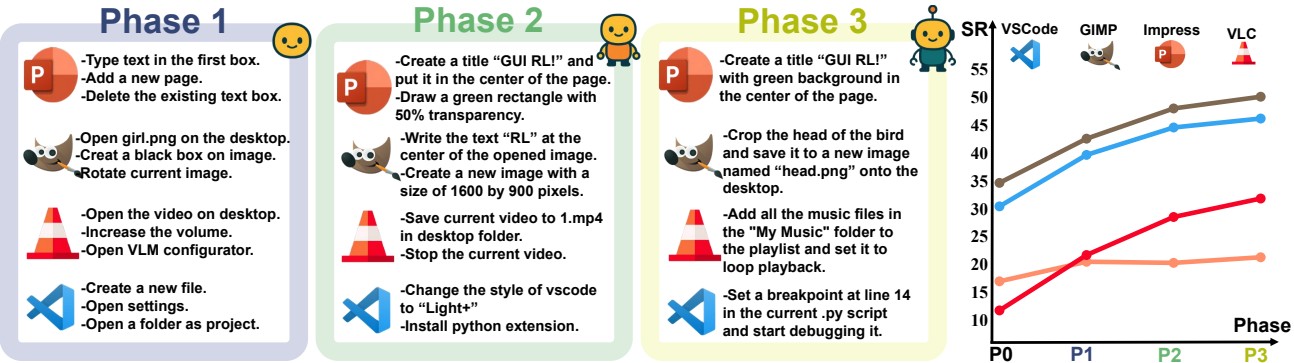

*Figure 4.* **Self-evolved task instructions and success rate (SR) curves across different software.** Tasks are progressively upgraded by the Curriculum Generator without human intervention, based on the evolving capabilities of the Actor Model at different training phases.

*Table 5.* **Step-wise vs. trajectory-level RL** under the ARPO (Lu et al., 2025) multi-turn setting on VSCode (success rate at different step budgets). Step-wise RL is preferable under the standard step-limited setting, while trajectory-level RL with critical-wrong-step masking gains an advantage as the interaction budget becomes large enough to exploit error recovery.

| Reward design | @5 | @10 | @15 | @25 | @50 |
|---|---|---|---|---|---|
| WSM trajectory (success/fail) | 8.2 | 14.1 | 23.5 | 25.9 | 30.6 |
| WSM trajectory + critical-step mask | 10.6 | 12.9 | 25.9 | 35.3 | **40.0** |
| WSM step-wise (ours) | **11.8** | **21.2** | **29.4** | 31.8 | 34.1 |

confirming that judgment precision rather than model scale is the primary bottleneck for self-evolution. Second, when reinforcement fine-tuning is enabled, GRPO outperforms SFT (behavior cloning), and adding adversarial imitation (AI) on failure steps further improves the policy by allowing it to learn from its own mistakes. The full system (World State Model + GRPO + AI) achieves the best result on every environment, with the largest absolute gain observed on VLC (11.8 → 31.8), an environment where the base model is initially weakest and most reliant on dense step-level feedback.

**Step-wise vs. Trajectory-level Optimization.** We further compare SEAgent against trajectory-level RL on VSCode under the ARPO (Lu et al., 2025) multi-turn protocol with different step budgets (Tab. 5). Two observations emerge. (i) Even under trajectory-level supervision, masking the steps before the first critical error already improves over a plain success/fail reward, which itself supports the value of WSM in providing finer-grained signals. (ii) Step-wise RL is preferable under the standard step-limited setting (@5–@15), because denser supervision yields more informative credit assignment; trajectory-level RL with critical-step masking only takes the lead at large step budgets (@25, @50), where the agent can afford to make mistakes and then recover. The two regimes are therefore complementary: step-wise RL is better aligned with OSWorld-style step-limited benchmarks, while trajectory-level RL can better exploit recovery behavior in more fault-tolerant deployments.

## 5. Conclusion

In this work, we introduce SEAgent, an autonomous Computer Use Agent (CUA) exploration system that learns from its own experience on specific software. Powered by a robust World State Model that provides step-level reward signals, and a carefully designed reinforcement learning framework that encourages free-form reasoning through trial and error, the CUA is able to evolve into a specialist for individual software platforms. Furthermore, a specialist-to-generalist training strategy enables the development of a strong generalist agent capable of operating across multiple software environments. Given that computer software constitutes a highly regularized virtual world, we believe this work can inspire future research on agentic systems in both gaming and real world embodied environments.

**Limitations and future work.** Our system has three remaining limitations. (1) SEAgent relies on the World State Model rather than real environment signals, since learning purely from sparse rewards in complex GUIs is still challenging; tighter integration with executable verifiers is a natural next step. (2) Although we evaluate on diverse and previously unseen software (e.g., LibreOffice tools, GIMP, ScienceBoard scientific applications), each task is still completable by a human in fewer than 20 steps, while real expert workflows can span hours. Scaling SEAgent to such long-horizon tasks is an interesting direction. (3) As shown in Tab. 5, step-wise RL is best suited to step-limited evaluation; in highly fault-tolerant deployments where the agent can afford to fail and recover, trajectory-level RL becomes complementary, and a hybrid that combines step-wise dense rewards with trajectory-level recovery is a promising future direction.

## Impact Statements

This paper presents work whose goal is to advance the field of machine learning. There are many potential societal consequences of our work, none of which we feel must be specifically highlighted here.

## Acknowledgments

This project is funded in part by Shanghai Artificial Intelligence Laboratory, Shanghai Innovation Institute, the Centre for Perceptual and Interactive Intelligence (CPII) Ltd under the Innovation and Technology Commission (ITC)'s InnoHK. Dahua Lin is a PI of CPII under the InnoHK.

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

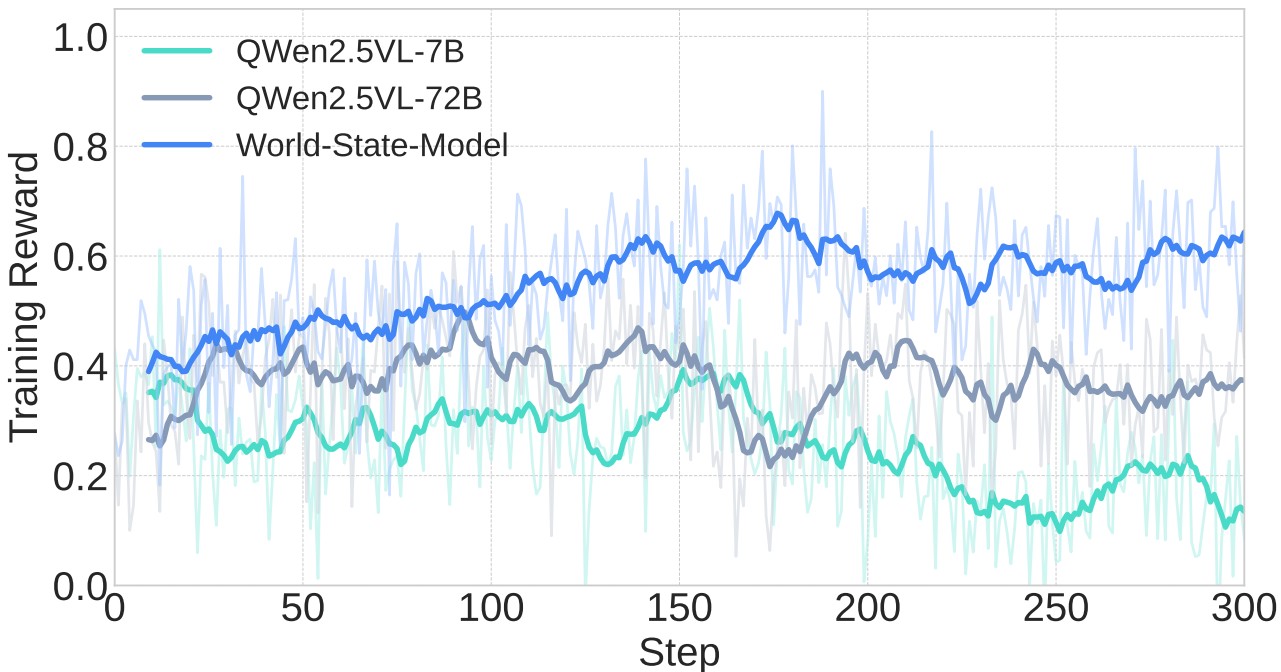

*Figure 5.* **Training reward with different reward signal provider.** Our World State Model provide reward signal that can achieve improved training reward compared to strong base models.

## A. World State Model

The World State Model (WSM) is a central component of SEAgent, responsible for understanding visual state changes and evaluating the effectiveness of the agent's actions.

### A.1. Model Architecture and Operation

The WSM is built upon the Qwen2.5-VL-7B vision-language model. It operates in two distinct modes, each with a specific input-output structure to perform different tasks:

1. **Trajectory Judgment:**

   **Input:** A sequence of screenshot images captured during an episode.

   **Output:** Short captions for each screenshot, the reasoning process for the judgment, and a structured judgment dictionary (containing fields such as `Correctness`, `Redundant`, and `First Error Step`, as detailed in Fig. 8 of the supplementary material).

2. **State Change Description:**

   **Input:** Two screenshot images, one from before and one after a single action was executed.

   **Output:** A detailed description of the visual differences between the two images.

### A.2. Fine-Tuning Dataset and Process

To equip the WSM with these capabilities, a specialized dataset was constructed for fine-tuning.

**Data Construction** The data construction process is as follows:

1. **Trajectory Sampling:** A Computer Using Agent (CUA), powered by UI-TARS and Gemini-2.5-Pro, was used to sample trajectories from 43 feasible tasks in Google Chrome within the OSWorld benchmark. These trajectories were saved as screenshot sequences.

2. **GPT-4o Annotation:** Using the prompts detailed in Figures 6 and 7 of the supplementary material, GPT-4o was employed to annotate the sampled trajectories, generating judgments and screenshot captions. Only samples where the judgment matched the ground truth from OSWorld evaluation protocols were retained, resulting in 860 high-quality annotated trajectories.

3. **Change Description Data:** An additional 1,000 pairs of (before action, after action) screenshots were sampled. GPT-4o was used to generate detailed descriptions of the differences, creating a 1,000-sample Change Description (CD) dataset.

**Fine-Tuning Process**  The fine-tuning was performed using the Llama-Factory framework on 8 NVIDIA A100 (80G) GPUs for 2,000 iterations. A learning rate of $2 \times 10^{-5}$ was used, and LoRA (rank=128) was employed for parameter-efficient fine-tuning. The 860 annotated trajectories serve as the core training data for teaching the model trajectory judgment, captioning, and reasoning. The 1,000-sample CD dataset acts as auxiliary data, specifically to encourage the model to focus on fine-grained visual differences, which enhances its overall state understanding. As shown in Table 1 of the main paper, incorporating CD data significantly boosts judgment performance. The two datasets were combined for training without any special re-weighting.

### A.3. Reward Generation from Trajectory Analysis

The trajectory judgment capability of the WSM is the core source of the reward signal for reinforcement learning. After an agent executes a full trajectory $\mathcal{H} = \{s_0, a_0, s_1, a_1, \ldots, s_{\text{final}}\}$, the WSM analyzes it and outputs a structured judgment. Based on this output, actions within the trajectory are dynamically labeled as either positive actions ($a_T$) or failure actions ($a_F$):

- **Fully Successful Trajectory:** If `Correctness` is 'True' and there are no `Redundant` steps, all actions $a$ in the trajectory are labeled as $a_T$.

- **Successful but Inefficient Trajectory:** If `Correctness` is 'True' but `Redundant` steps begin at step $k$, all actions prior to step $k$ are labeled as $a_T$.

- **Failed Trajectory:** If `Correctness` is 'False' and the `First Error Step` is $e$, all actions prior to step $e$ are labeled as $a_T$, while the erroneous action $a_e$ is labeled as $a_F$.

These dynamically labeled $a_T$ and $a_F$ actions constitute the reward signals for the RL pipeline. During training, the actor predicts an action $a_t$ based on the history $\{a_0, s_0, \ldots, s_t\}$ and uses these labels to calculate rewards.

## B. Curriculum Generator

The Curriculum Generator is designed to dynamically produce tasks of increasing difficulty and diversity, guiding the agent through a systematic exploration of the software's capabilities.

### B.1. Task Generation Mechanism

The workflow of the Curriculum Generator is detailed in the pseudocode in our supplementary material. Its core idea is to leverage the WSM's analysis of completed tasks to generate new ones. The process, illustrated by the "add a rectangle" example from Figure 5, involves three main steps:

1. **Analysis and Feedback:** The agent successfully completes an initial task, "add a rectangle." The WSM analyzes the execution trajectory and extracts two key pieces of information: a task evaluation (`Exam`) and a list of observed state changes (`CD_list`).

    `CD_list`: {"add a rectangle": ["The Edit bar is expanded...", "The cursor has changed into a cross...", "A blue box appears on the screen with side bars showing properties such as fill, line, color, width, transparency, and corner style..."], ...}
    `Exam`: [{"task": "add a rectangle", "status": "success"}, ...]

2. **Knowledge Integration and Task Generation:** The `CD_list` and `Exam` are fed into the Curriculum Generator. It distills new knowledge, such as "properties of a rectangle," and integrates it into its internal `Software guidebook`. Based on this new knowledge, it generates more challenging tasks like "Add a green rectangle" or "Add a red rectangle with 50% transparency," which are then added to the task buffer.

3. **Iterative Learning:** In the next RL phase, the agent samples from this updated, more challenging task buffer. The continuously enriched `Software guidebook` acts as the system's long-term memory, driving the Curriculum Generator to propose increasingly sophisticated and unexplored tasks in subsequent rounds, thereby guiding the agent toward mastery.

## C. Details of Curriculum Generator.

### C.1. Exemplar Case during Task Evolution.

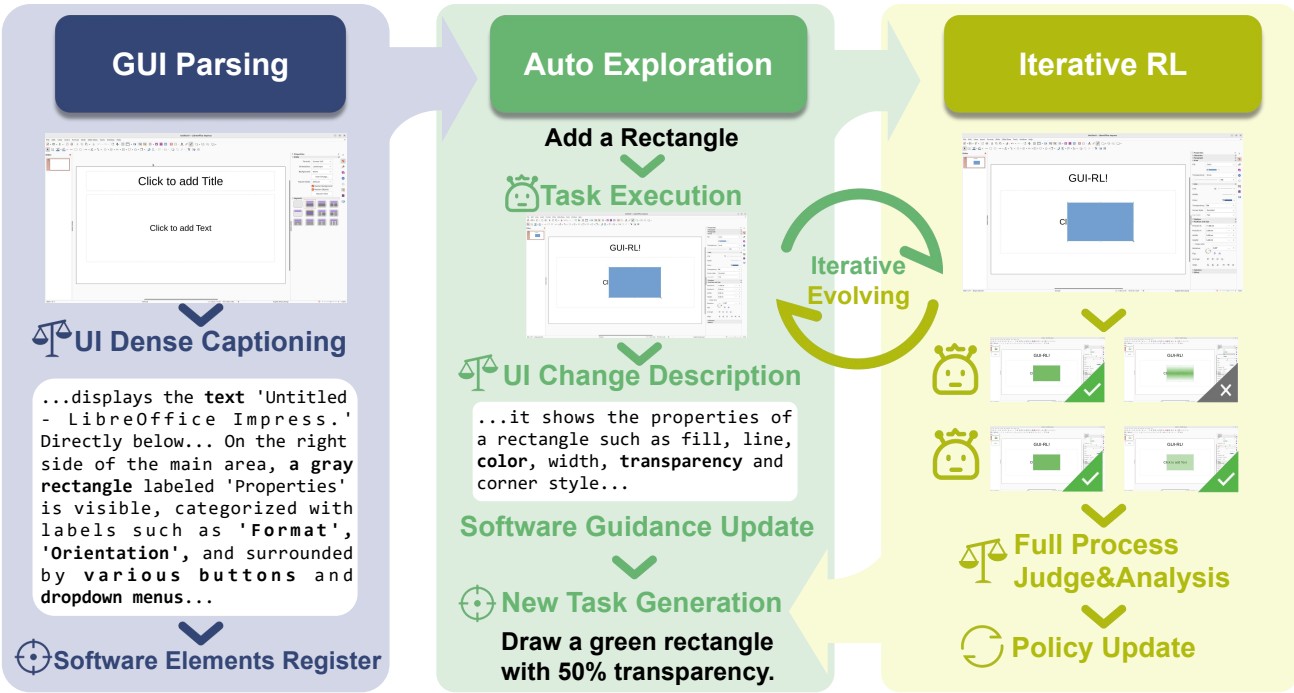

*Figure 6.* **SEAgent autonomous exploration pipeline.** The agent (policy model) and World State Model iteratively generate new task and perform RL to become a specialist in novel software.

We provide an exemplar case of our task evolution pipeline in Fig. 6, demonstrated using LibreOffice Impress. Initially, the World State Model parses a screenshot of the Impress interface into detailed captions describing the layout and individual buttons. The Task Generator then produces an initial task set, $\mathcal{I}_0 = \{I_0^{(1)}, I_0^{(2)}, \ldots\}$, and summarizes the initial software guidance memory $U_0$. The initial agent executes tasks in $\mathcal{I}_0$, such as "Add a Rectangle," while the World State Model evaluates these actions, providing judgments and detailed descriptions of resulting changes. As shown in the Auto-Exploration stage, this includes generating captions for newly appeared property panels and assessing execution success. The Task Generator incorporates feedback on execution success and newly revealed properties (e.g., transparency) to evolve new tasks, such as "Draw a green rectangle with 50% transparency." This process iteratively improves through reinforcement learning, enabling continuous task evolution and agent self-improvement.

### C.2. Comparative Analysis of Instruction Generation Strategies.

To validate the effectiveness of our Curriculum Generator, we conducted a comparative analysis against state-of-the-art instruction generation methods, namely those from NNetNav (Murty et al., 2025) and WebRL (Qi et al., 2024).

**Experimental Setup**   We adapted the official code and prompts from these prior works from web environments to general software applications. To ensure a fair comparison of the curriculum quality, for each strategy, we employed two leading LLMs: the open-source Qwen2.5-72B (Bai et al., 2025) and the proprietary Gemini-2.5-Pro (Google DeepMind, 2025b). The tasks generated by each strategy were used to train an RL agent (using GRPO only), with reward signals uniformly provided by our fine-tuned WSM. The evaluation was performed on two applications: VSCode from OSWorld (a standard software) and Celestia from ScienceBoard (Sun et al., 2025) (a more challenging, out-of-domain scientific application). The primary metric was the task success rate.

*Table 6.* Success rate (%) comparison of different task generation strategies on two software applications.

| Task Generation Strategy | LLM | VSCode | Celestia |
|---|---|---|---|
| WebRL | Qwen2.5-72B | 27.5 | 0.00 |
| WebRL | Gemini2.5-Pro-thinking | 36.2 | 3.03 |
| NNetNav | Qwen2.5-72B | 34.6 | 0.00 |
| NNetNav | Gemini2.5-Pro-thinking | 43.6 | 5.05 |
| Curriculum Generator (Ours) | Qwen2.5-72B | 37.7 | 9.09 |
| Curriculum Generator (Ours) | Gemini2.5-Pro-thinking | 42.3 | 12.12 |

**Results and Discussion**   The results are presented in Table 6. As shown, the reverse instruction generation strategy from NNetNav (Murty et al., 2025) is highly effective on the in-domain application (VSCode), demonstrating high data generation efficiency by producing successful trajectories. However, a critical trade-off was observed: this approach tends to generate many similar tasks, limiting its ability to explore the full breadth of the software's functionalities. This limitation becomes more pronounced when the task generator is unfamiliar with the target software, as seen in the OOD Celestia environment.

In contrast, our guidebook-based method, while having a lower initial data generation efficiency, excels at systematic exploration. It builds structured knowledge of the software from scratch, making it more robust for tackling novel applications. This is evidenced by its superior performance on the more challenging Celestia software.

We conclude that these two strategies are complementary. Reverse instruction generation can efficiently exploit known functionalities, while our guidebook-based method can systematically explore new ones and help the task generator build a more comprehensive understanding of the target software. A hybrid approach combining both strategies is a promising direction for future work.

## D. Additional Analysis from the Rebuttal

**Single-model self-evolution.**   The main paper uses a stronger LLM (Qwen2.5-72B) as the Curriculum Generator together with UI-TARS as the actor. A natural question is whether a single, sufficiently capable model can self-evolve without an external task generator. We probe this by running our pipeline with a single backbone serving as both the actor and the task generator. Results on VSCode are summarized below:

| Model | Before | After self-evolve |
|---|---|---|
| Qwen2.5-VL-7B | 6.1 | 9.6 |
| Qwen2.5-VL-32B | 10.4 | 16.5 |
| Qwen3-VL-8B (Qwen Team, 2025) | 43.7 | 49.5 |

For weak base models (Qwen2.5-VL-7B/32B) the gain is modest, which justifies the choice in the main paper to combine a CUA-specialized actor (UI-TARS) with a stronger open-source LLM. As the base model becomes stronger (Qwen3-VL-8B), single-model self-evolution becomes effective on its own. This indicates that the term "self-evolving" in this paper refers to the closed-loop, human-free training pipeline rather than to a constraint that a single model improves entirely on its own.

**Applying World State Model dense reward to other RL pipelines.**   To isolate the contribution of the reward model from that of the optimization algorithm, we additionally plug the dense reward signal of World State Model into DigiRL (Bai et al., 2024) and WebRL (Qi et al., 2024). The results are reported below:

| Method | VSCode | GIMP | Impress |
|---|---|---|---|
| DigiRL (Bai et al., 2024) | 43.7 | 45.4 | 19.6 |
| WebRL (Qi et al., 2024) | 36.5 | 37.7 | 20.4 |
| DigiRL + World State Model dense reward | 44.3 | 47.7 | 20.4 |
| WebRL + World State Model dense reward | 39.4 | 39.5 | 21.3 |
| GRPO + World State Model dense reward | 41.7 | 41.5 | 20.9 |
| SEAgent (AI + GRPO + World State Model) | **46.1** | **50.0** | **21.3** |

The World State Model dense reward improves both DigiRL and WebRL on most settings, confirming that the reward model is a generally useful component. The full SEAgent remains the strongest overall.

**Diversity of the Curriculum Generator.** Tab. 6 measures task quality through downstream success rate. To complement this with a more direct measure of the generated tasks themselves, we additionally evaluate semantic diversity. For each task generator, we encode the same number of generated task descriptions using OpenAI `text-embedding-3-small` and compute (i) the normalized embedding variance ($\text{Var}_{\text{norm}}$), capturing global semantic spread, and (ii) the normalized average pairwise cosine distance ($\text{Dist}_{\text{norm}}$), capturing local semantic dissimilarity. The diversity score is $0.5\,\text{Var}_{\text{norm}} + 0.5\,\text{Dist}_{\text{norm}}$.

| Task generator | $\text{Var}_{\text{norm}}$ | $\text{Dist}_{\text{norm}}$ | Diversity |
|---|---|---|---|
| WebRL (Qwen2.5-72B) | 0.36 | 0.40 | 0.38 |
| WebRL (Gemini-2.5-Pro-Thinking) | 0.61 | 0.71 | 0.66 |
| NNetNav (Qwen2.5-72B) | 0.48 | 0.50 | 0.49 |
| NNetNav (Gemini-2.5-Pro-Thinking) | 0.70 | 0.64 | 0.67 |
| Ours (Qwen2.5-72B) | 0.65 | 0.67 | 0.66 |
| Ours (Gemini-2.5-Pro-Thinking) | **0.78** | **0.80** | **0.79** |

Our curriculum generator yields higher diversity than WebRL and NNetNav under the same backbone. The trend is consistent across both Qwen2.5-72B (used in the main paper for an open-source pipeline) and Gemini-2.5-Pro-Thinking.

**Ablation on guidebook updates.** We isolate the contribution of the guidebook-update mechanism in the Curriculum Generator. Removing the guidebook update has a small effect on familiar software (VSCode, GIMP) but a large effect on more novel software (VLC):

| Setting | VSCode | GIMP | VLC |
|---|---|---|---|
| w/o guidebook update | 44.3 | 46.2 | 13.4 |
| w/ guidebook update | **46.1** | **50.0** | **31.8** |

We hypothesize this gap is because the open-source generator (Qwen2.5-72B) is already familiar with widely used software like VSCode and GIMP and can propose meaningful tasks even without the guidebook, while VLC is much less common in pretraining data; in this case the guidebook helps the system build a structured understanding of the interface and produce more suitable tasks.

# E. Experiments on AndroidWorld

*Table 7.* Success Rate on AndroidWorld (Rawles et al., 2024)

| Model | AndroidWorld_SR |
|---|---|
| Qwen2.5-VL-7B | 8.0 |
| Qwen2.5-VL-7B+SEAgent | 19.5 |
| UI-TARS-7B-SFT | 33.0 |
| UI-TARS-7B-SFT+SEAgent | 38.0 |

To evaluate SEAgent's application to other formats of GUI, we conduct experiments on the AndroidWorld (Rawles et al., 2024) benchmark, which focuses on mobile GUIs. We apply our SEAgent pipeline to two distinct backbone models.

As shown in the table above, our method yields substantial performance improvements for both, demonstrating that its self-evolving approach is effective across different model architectures and GUI formats. Specifically, SEAgent improves the success rate of Qwen2.5-VL by $+11.5\%$ and UI-TARS by $+5.0\%$. This indicates that our pipeline also generalizes to other forms of GUI.

We additionally evaluate World State Model, which was distilled only from $860$ Chrome-based desktop trajectories, on AndroidWorld for trajectory judgment without any further finetuning:

| Setting | Precision | NPV |
|---|---|---|
| Before World State Model training (Qwen2.5-VL-7B) | 41.9 | 79.0 |
| After World State Model training | **58.4** | **84.1** |

The improvement on AndroidWorld is consistent with the cross-domain transfer observed in Tab. 1, indicating that the judgment ability acquired by World State Model generalizes from desktop to mobile GUIs.

## F. Sensitivity Analysis on Key Hyperparameters

We conducted a sensitivity analysis on key hyperparameters to evaluate their impact on the SEAgent pipeline. For model sampling, we set the temperature $t = 0$ for better reproducibility. We analyze two specific parameters: the number of generated tasks and the number of change descriptions. The results are presented in Table 8 and discussed below.

*Table 8.* Sensitivity analysis for key hyperparameters in the SEAgent pipeline, evaluated on VSCode. The metric is Success Rate (%).

| # Tasks Generated | VScode SR | # Change Descriptions | VScode SR |
|---|---|---|---|
| 30 | 31.88 | 30 | 33.33 |
| 50 | 36.23 | 50 | 37.68 |
| 100 | 37.68 | 100 | 37.68 |
| 200 | 37.68 | 200 | 34.78 |

**Number of Generated Tasks**    This parameter controls the breadth of exploration in each learning cycle. As shown in our analysis, performance improves as more diverse tasks are generated, eventually plateauing around 100 tasks.

**Number of Change Descriptions**    This parameter controls how much new information the generator receives to update its "software guidebook." We found a clear trade-off: A sufficient number of descriptions (50–100) is essential for the generator to learn about new UI functionalities and create meaningful, unexplored tasks. However, providing too many descriptions (e.g., 200) creates an overly long context for the LLM, which degrades the quality of task generation and hurts final performance.

## G. Ablation on the Loss Balance Factor.

In Sec.3.2, we use $\gamma$ to balance the ratio of two loss item: adversarial imitation that learn from error and GRPO that learn to achieve success. We ablate the choice of $\gamma$ in Tab.9, according to which we set $\gamma = 0.2$ in main experiments.

*Table 9.* VScode Success Rate on OSWorld (Xie et al., 2024) under different loss balance factor $\gamma$ values.

| $\gamma$ | 0.0 | 0.1 | 0.2 | 0.3 | 0.5 | 0.8 |
|---|---|---|---|---|---|---|
| Success Rate (%) | 34.8 | 36.2 | 37.7 | 31.9 | 26.1 | 23.1 |

## H. Reward Function for Different Actions.

As listed in Tab. 10

*Table 10.* Reward computation for each action type in GUI agent

| Action Type | Description | Distance-based Reward |
|---|---|---|
| `click`, `left_single`, `right_single`, `hover` | Click or hover on a location | Normalized L1 distance between predicted and ground-truth coordinates |
| `left_double`, `double_click` | Double click on a region | Normalized L1 distance between clicked coordinates |
| `drag`, `select` | Drag from start box to end box | Intersection over Union (IoU) between predicted and ground-truth boxes |
| `type` | Type textual input | Character-level BLEU score between predicted and ground-truth text |
| `hotkey` | Press multiple keys at once | Character-level BLEU score between predicted and ground-truth key combinations |
| `press` | Press a single key | Character-level BLEU score between predicted and ground-truth key |
| `scroll` | Scroll in a certain direction | Character-level BLEU score between predicted and ground-truth direction |
| `move_mouse` | Move mouse to a specific location | Normalized L1 distance between predicted and ground-truth coordinates |
| `highlight` | Highlight a rectangular UI region | IoU between predicted and ground-truth region |
| `copy`, `paste` | Clipboard operations | BLEU score between copied/pasted content |
| `wait` | Explicit wait command | Fixed reward + 1 |
| `finished`, `finish_task` | Finish current task/trajectory | Fixed reward + 1 |

## I. Data Statistics during Iterative Reinforcement Learning.

As listed in Tab. 11

*Table 11.* Number of episode (Success/Failure) across four phases for different software tools during self-evolution. Each episode contains 8.8 multi-turn conversions in average.

|  | Phase0 | Phase1 | Phase2 | Phase3 |
|---|---|---|---|---|
| VSCode | 112/39 | 282/83 | 161/34 | 98/55 |
| GIMP | 104/51 | 309/90 | 183/50 | 95/52 |
| Impress | 102/44 | 290/92 | 185/61 | 87/51 |
| VLC | 85/29 | 114/41 | 160/48 | 53/27 |
| Writer | 123/62 | 278/101 | 201/69 | 101/43 |

## J. Detailed Prompt Templates.

For evaluation on AgentRewardBench (Lù et al., 2025), we use their official template for final state screenshot only testing and modified prompt in Fig.7 for entire process (or sampled middle screenshots) testing.

For evaluation on OSWorld Sampled trajectories, we use prompt in Fig.8 to prompt GPT-4o to provide step level judges, the sampled judges on Chrome in OSWorld (Xie et al., 2024) serves as training data of GUI-Judge. This template is also used in training GUI-Judge and at inference time in autonomous exploration stage.

For navigator, we use prompt template in Fig.9, which takes previous software usage manual and the performance of actor agent evaluated by judge (Empty if in initial phase.) as well as detailed exploration caption as input and output the updated usage manual as well as new task for agent to execute.

## K. Self documented usage manual on different software during exploration.

In Fig.10 Fig.12, Fig.11, Fig.13, we demonstrate the self-documented usage manuals of the navigator (Qwen2.5-72B (Yang et al., 2024)) in the exploration and learning system introduced in Sec.3.1.

## L. Broader Impacts

**Potential positive societal impacts:** SEAgent introduces a self-evolving paradigm for Computer Use Agents (CUAs), enabling them to autonomously learn and adapt to previously unseen software without human supervision. This significantly reduces the need for extensive manual data annotation and domain-specific customization, allowing intelligent agents to assist users across a wide range of applications—including productivity tools, multimedia editing, and educational software. By automating repetitive tasks and providing guidance in complex software environments, SEAgent holds promise for improving accessibility, enhancing digital literacy, and reducing cognitive workload in both professional and everyday settings.

**Potential negative societal impacts:** The capability of SEAgent to autonomously explore and operate complex software also introduces risks of misuse. Malicious actors might repurpose SEAgent for unauthorized software automation, such as automating account creation, spamming interfaces, or conducting surveillance via GUI interactions. In addition, as the agent learns from its own experience, there exists a risk that the agent may inadvertently inherit or amplify software-specific biases, potentially leading to unfair or inappropriate behaviors in sensitive applications (e.g., finance, legal automation). Mitigation strategies include controlled release of models, behavior filters during deployment, and incorporating safeguards in the World State Model to detect and prevent unintended or adversarial behavior.

---

**Algorithm 1** SEAgent Specialized Self-Evolution Training Loop

---

1: **Input:** Initial policy $\pi_0$, World State Model $\mathcal{M}_{\text{state}}$, Curriculum Generator $\mathcal{M}_{\text{task}}$, Initial GUI state $S_0$
2: **1. Task Initialization**
3: $\mathcal{C}_0 \leftarrow \text{CaptionGUI}(S_0)$                                               ▷ Parse initial GUI layout (menu bar, buttons, etc.)
4: $\mathcal{I}_0, U_0 \leftarrow \mathcal{M}_{\text{task}}(\emptyset, \emptyset, \emptyset, \mathcal{C}_0)$                                ▷ Generate basic initial tasks and usage guide
5: **2. Self-Evolution Phase Loop:**
6: **for** $p = 0$ to $P - 1$ **do**
7:     **2.1 Autonomous Exploration**
8:     $\mathcal{D}_{\text{traj}} \leftarrow \emptyset$
9:     **for all** $I \in \mathcal{I}_p$ **do**
10:         $\tau \leftarrow \text{ExecuteInstruction}(\pi_p, I)$                       ▷ Actor executes task in the virtual environment
11:         **2.2 Effect Evaluation**
12:         $\mathcal{J}_I, \mathcal{C}_I \leftarrow \mathcal{M}_{\text{state}}(\tau)$                  ▷ Step-level trajectory judgment and new state captions
13:         $\mathcal{D}_{\text{traj}} \leftarrow \mathcal{D}_{\text{traj}} \cup \{(\tau, \mathcal{J}_I, \mathcal{C}_I)\}$      ▷ $\mathcal{J}_I$: a sequence of per-step feedback labels ($a_T$ or $a_F$)
14:     **end for**
15:     **2.3 Policy Update (RFT)**
16:     Split $\mathcal{D}_{\text{traj}}$ into:
17:         $\mathcal{D}_{\text{pos}}$: steps labeled as positive $a_T$
18:         $\mathcal{D}_{\text{neg}}$: steps labeled as negative $a_F$
19:     Compute GRPO loss on $\mathcal{D}_{\text{pos}}$:
20:         $r(a, a_T) = \mathbb{I}[\text{type}(a) = \text{type}(a_T)] + r_{\text{dist}}(a, a_T)$
21:     Compute Adversarial Imitation loss on $\mathcal{D}_{\text{neg}}$:
22:         $\mathcal{L}_{\text{AI}} = -\log \frac{\pi_\theta(a|s, I)}{\pi_{\text{ref}}(a_F|s, I)}$
23:     Total loss: $\mathcal{L}_{\text{total}} = \mathcal{L}_{\text{GRPO}} + \gamma \mathcal{L}_{\text{AI}}$
24:     $\pi_{p+1} \leftarrow \text{Update}(\pi_p, \mathcal{L}_{\text{total}})$
25:     **2.4 Task Update**
26:     $\mathcal{I}_{p+1}, U_{p+1} \leftarrow \mathcal{M}_{\text{task}}(U_p, \mathcal{I}_p, \{\mathcal{J}_I\}, \{\mathcal{C}_I\})$    ▷ Generate more complex tasks based on new software knowledge and performance feedback
27: **end for**
28: **Output:** Specialized agent policy $\pi_P$ after $P$ stages of self-evolution

---

# M. SEAgent Self-Evolution Algorithm

Algorithm 1 presents the core self-evolution training loop of SEAgent in a specialized software environment. The procedure is divided into four major stages:

(1) **Task Initialization.** Given the initial GUI state of a target software application, the World State Model performs dense captioning to extract structural semantics (e.g., menu bar, buttons), which is used by the Curriculum Generator to create an initial set of executable tasks and an editable software guidebook.

(2) **Autonomous Exploration and Effect Evaluation.** The agent explores each task via its current policy. The World State Model then performs step-level trajectory analysis, assigning each action a feedback label—either correct ($a_T$) or incorrect ($a_F$)—and generating GUI state change captions. This produces rich supervision signals for both policy learning and downstream task generation.

(3) **Policy Update via Reinforcement Fine-Tuning.** Based on the labeled execution data, positive and negative action steps

are separated. We apply Group Relative Policy Optimization (GRPO) to reinforce correct actions, and Adversarial Imitation (AI) to suppress failure-prone behaviors. The updated policy is used for the next exploration round.

(4) **Task Update.** The Curriculum Generator leverages feedback signals ($\mathcal{J}$) and GUI state transitions ($\mathcal{C}$) to propose more diverse and challenging tasks, thereby expanding the task frontier in a curriculum fashion.

This process repeats over multiple curriculum phases, ultimately yielding a specialized agent policy capable of mastering complex operations in the given software environment.

## Web Step Level Judge Prompt Template

You are a Language Model specialized in judging the performance of web agents in web-navigation tasks. For a certain website, you are given the goal of a navigation task, the current URL of the webpage, the actions taken by the agent, and the thought process of the agent. **Additionally, you will have access to the sequence of key frame screenshots** Your task is to answer several questions about the agent's performance in the task.
**You should carefully look at the sequencial screenshot images in order to decide whether its sucessfully finish the task or failed halfway.**

Question 1: Was the sequence of actions successful in achieving the goal?
Choices: <success>Successful</success>, <success>Unsuccessful</success>

Question 2: Did the agent perform unnecessary actions that could lead to unintended side effects?
Choices: <side>Yes</side>, <side>No</side>

Question 3: Did the agent perform the task optimally, by only performing necessary actions and avoiding unnecessary ones?
Choices:
<optimal>1. Complete Failure</optimal>
<optimal>2. Suboptimal</optimal>
<optimal>3. Somewhat Optimal</optimal>
<optimal>4. Completely Optimal</optimal>

Question 4: Did the agent loop through a sequence of actions that did not make progress towards the goal?
Choices: <loop>Yes</loop>, <loop>No</loop>

Provide your reasoning for each question.
Your answer **must** follow this exact format:

<reasoning>your reasoning here</reasoning>
<success>answer</success>
<side>answer</side>
<optimal>answer</optimal>
<loop>answer</loop>

*Figure 7.* **Prompt Template of GUI-Judge for web agent trajectories evaluations** with history screenshots as input, its difference with default prompt of AgentRewardBench (Lù et al., 2025) is highlighted in bold.

## OSWorld Step Level Judge Prompt Template

I am evaluating the performance of a UI agent. The images provided are sequential keyframes that represent the full execution trajectory of the agent when attempting to follow a command. These keyframes correspond to the instruction: [INSTRUCTION].

Please thoroughly analyze the sequence to assess the following aspects:

1. Correctness — Did the agent successfully complete the task as instructed?
2. Redundant Steps — Identify any unnecessary or repeated actions that do not contribute to the goal.
3. Optimization — Did the agent follow an efficient plan with a minimal number of steps?
5. First Error Step — If the execution is incorrect or sub-optimal, determine the index of the first 5. keyframe where a mistake occurred.
6. Error Analysis — Provide a brief explanation of the mistake at that step.
7. Correct Action Suggestion — Explain what the agent should have done instead at the point of error.

Important Instructions:
The agent may have made progress toward the goal, but unless the task is fully and correctly completed, you must set 'Correctness' to False.
Be cautious in determining success. Missing confirmation screens, skipped inputs, or wrong UI elements clicked all count as errors.
Carefully examine all UI changes, button interactions, text entries, and any visual feedback in the screenshots.
Clearly indicate which exact steps are redundant (starting from 1).
Once you finish the analysis, return your evaluation in the following dictionary format. Include your step-by-step reasoning above the result.

```
<thinking>step by step reasoning.</thinking>
res_dict = {
    "Correctness": True or False,
    "Redundant": [step numbers],
    "Optimized": True or False,
    "First_Error_Step": step number or None,
    "Error_Type": "brief description of the mistake",
    "Correct_Action": "what should have been done instead"
}
```

*Figure 8.* **Prompt Template of GUI-Judge for OSWorld (Xie et al., 2024) trajectories**, which prompts judge model to provide step level reward signal.

## Task Buffer Update Prompt Template

You are now a teacher training a Computer Use Agent (CUA). This CUA is exposed to a new software environment and undergoes multiple rounds of iterative training. Your task is to issue new tasks for the agent to explore and train on, based on the feedback from the agent's actions. You are also responsible for summarizing a software usage manual to help the agent remember knowledge about the software.

The agent has provided the following feedback on its operations within the software: {json.dumps(action_decription_list)}

Here is the software usage document you summarized in the previous round: {document}

Here is the agent's performance on the task you provided in the previous round: {json.dumps(exam)}

Your are also access to the previous given tasks with the screenshot caption after agent's execution. You can also use these captions and results to evaluate the agent's capability and generate new task and update document accordingly given the caption of the new screen and the corresponding intruction with judged evaluation: {json.dumps(prev_states)}

Please:
- Analyze the agent's performance.
- Integrate new knowledge from the feedback.
- Update the usage manual accordingly.
- Design a new set of tasks (with increased difficulty) (30 or more) that reinforce the concepts the agent struggled with in the last round.
- Each task **must be concise and specific**, targeting a concrete atomic action, based on the document and agent's observations, such as:
    - "Create a file named main.py."
    - "Open Terminal card."
- Each task must be executable from software initial state with no file open, e.g. you should not generate task like save xxx.txt if xxx.txt doesn't exist or created.
- if task is in sequential order with reliance, you should output a seq list like [subtask1, subtask2, ...], if there is no reliance, output [task].
- Decompose and target previous errors in a more focused way.

Output your reasoning and analysis process first. Then output the updated usage document and task list in the following JSON format within a SINGLE JSON DICT easier for me to parse:

json
{{
    "software_document_new": "...",
    "exam_new": [[subtask1, subtask2, ...], [task]...]
}}

*Figure 9.* **Prompt Template for task buffer update**, which generates new tasks in a curriculum manner and update software documents. The new tasks are used for actor to perform next phase of RL.

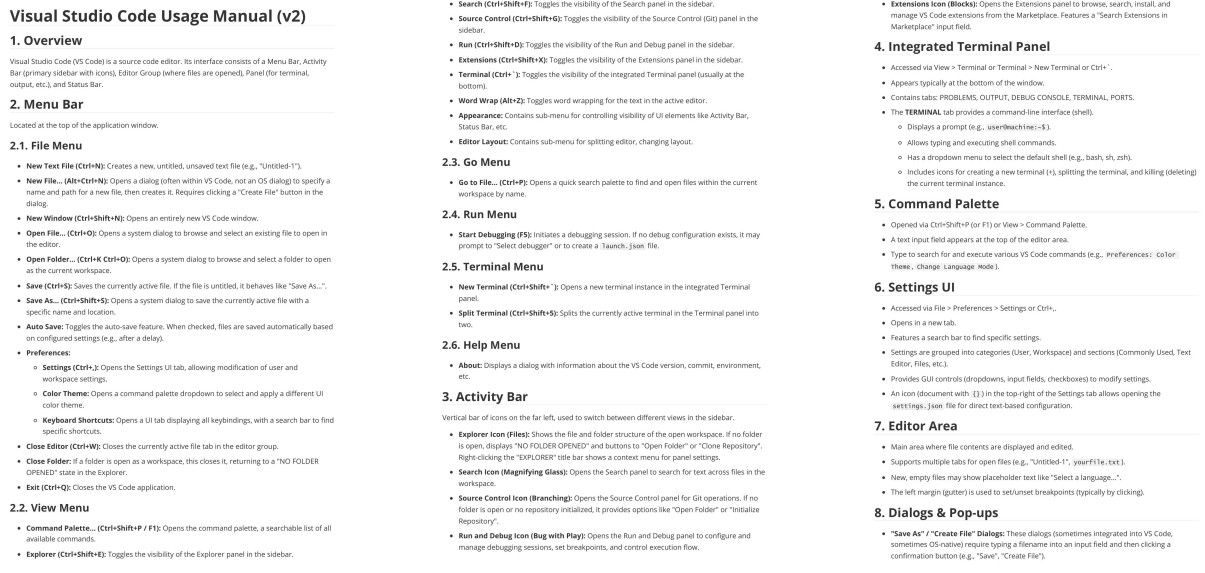

*Figure 10.* **Automatically generated usage manual during self exploration** on VScode.

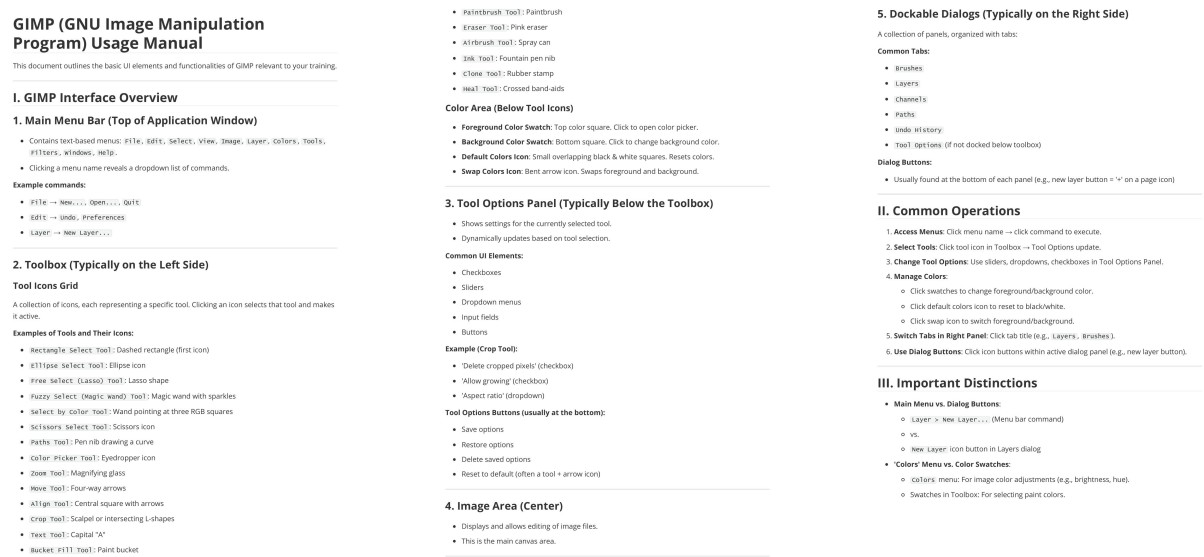

*Figure 11.* **Automatically generated usage manual during self exploration** on GIMP.

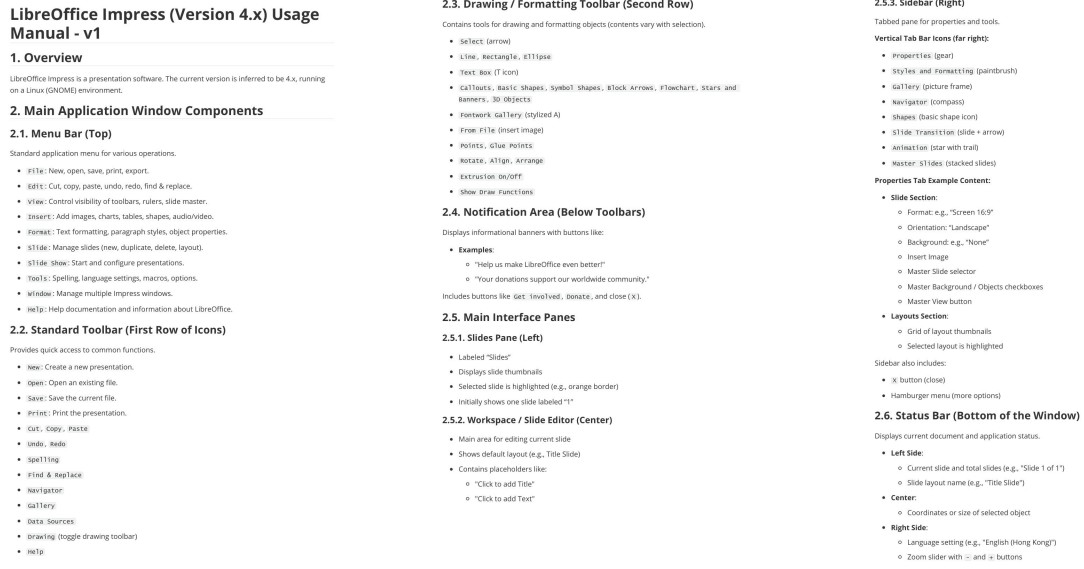

*Figure 12.* **Automatically generated usage manual during self exploration** on LibreOffice_Impress.

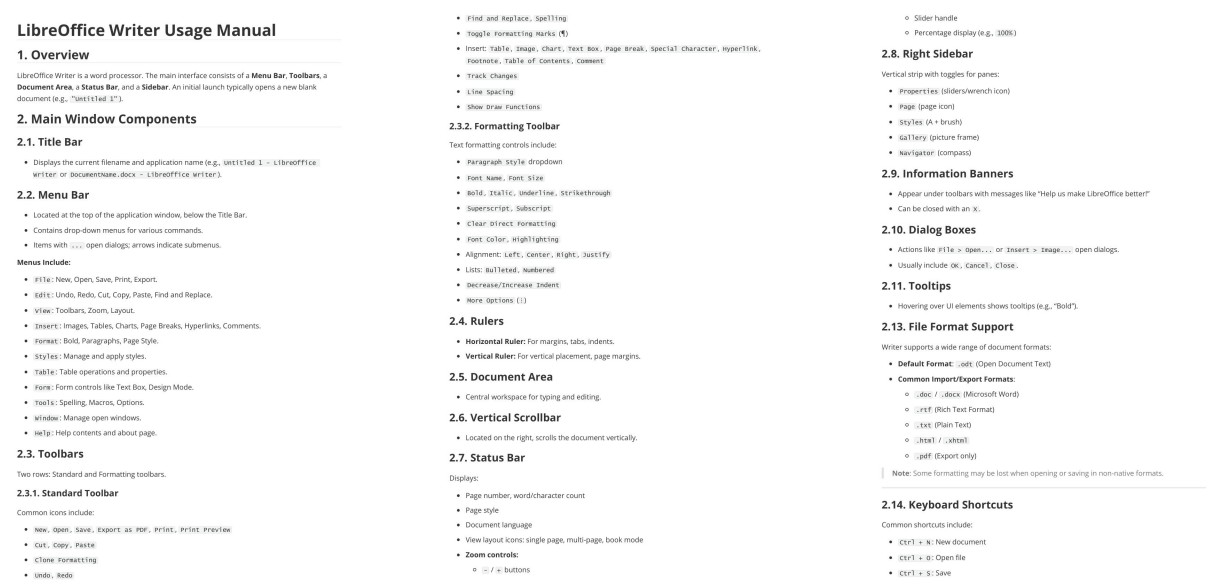

*Figure 13.* **Automatically generated usage manual during self exploration** on LibreOffice_Writer.

