**Visual Studio Code Usage Manual (v2)**

**1. Overview**

Visual Studio Code (VS Code) is a source code editor. Its interface consists of a Menu Bar, Activity Bar (primary sidebar with icons), Editor Group (where files are opened), Panel (for terminal, output, etc.), and Status Bar.

**2. Menu Bar**

Located at the top of the application window.

**2.1. File Menu**

- **New Text File (Ctrl+N):** Creates a new, untitled, unsaved text file (e.g., "Untitled-1").
- **New File... (Alt+Ctrl+N):** Opens a dialog (often within VS Code, not an OS dialog) to specify a name and path for a new file, then creates it. Requires clicking a "Create File" button in the dialog.
- **New Window (Ctrl+Shift+N):** Opens an entirely new VS Code window.
- **Open File... (Ctrl+O):** Opens a system dialog to browse and select an existing file to open in the editor.
- **Open Folder... (Ctrl+K Ctrl+O):** Opens a system dialog to browse and select a folder to open as the current workspace.
- **Save (Ctrl+S):** Saves the currently active file. If the file is untitled, it behaves like "Save As...".
- **Save As... (Ctrl+Shift+S):** Opens a system dialog to save the currently active file with a specific name and location.
- **Auto Save:** Toggles the auto-save feature. When checked, files are saved automatically based on configured settings (e.g., after a delay).
- **Preferences:**
  - **Settings (Ctrl+,):** Opens the Settings UI tab, allowing modification of user and workspace settings.
  - **Color Theme:** Opens a command palette dropdown to select and apply a different UI color theme.
  - **Keyboard Shortcuts:** Opens a UI tab displaying all keybindings, with a search bar to find specific shortcuts.
- **Close Editor (Ctrl+W):** Closes the currently active file tab in the editor group.
- **Close Folder:** If a folder is open as a workspace, this closes it, returning to a "NO FOLDER OPENED" state in the Explorer.
- **Exit (Ctrl+Q):** Closes the VS Code application.

**2.2. View Menu**

- **Command Palette... (Ctrl+Shift+P / F1):** Opens the command palette, a searchable list of all available commands.
- **Explorer (Ctrl+Shift+E):** Toggles the visibility of the Explorer panel in the sidebar.

- **Search (Ctrl+Shift+F):** Toggles the visibility of the Search panel in the sidebar.
- **Source Control (Ctrl+Shift+G):** Toggles the visibility of the Source Control (Git) panel in the sidebar.
- **Run (Ctrl+Shift+D):** Toggles the visibility of the Run and Debug panel in the sidebar.
- **Extensions (Ctrl+Shift+X):** Toggles the visibility of the Extensions panel in the sidebar.
- **Terminal (Ctrl+`):** Toggles the visibility of the integrated Terminal panel (usually at the bottom).
- **Word Wrap (Alt+Z):** Toggles word wrapping for the text in the active editor.
- **Appearance:** Contains sub-menu for controlling visibility of UI elements like Activity Bar, Status Bar, etc.
- **Editor Layout:** Contains sub-menu for splitting editor, changing layout.

**2.3. Go Menu**

- **Go to File... (Ctrl+P):** Opens a quick search palette to find and open files within the current workspace by name.

**2.4. Run Menu**

- **Start Debugging (F5):** Initiates a debugging session. If no debug configuration exists, it may prompt to "Select debugger" or to create a `launch.json` file.

**2.5. Terminal Menu**

- **New Terminal (Ctrl+Shift+`):** Opens a new terminal instance in the integrated Terminal panel.
- **Split Terminal (Ctrl+Shift+5):** Splits the currently active terminal in the Terminal panel into two.

**2.6. Help Menu**

- **About:** Displays a dialog with information about the VS Code version, commit, environment, etc.

**3. Activity Bar**

Vertical bar of icons on the far left, used to switch between different views in the sidebar.

- **Explorer Icon (Files):** Shows the file and folder structure of the open workspace. If no folder is open, displays "NO FOLDER OPENED" and buttons to "Open Folder" or "Clone Repository". Right-clicking the "EXPLORER" title bar shows a context menu for panel settings.
- **Search Icon (Magnifying Glass):** Opens the Search panel to search for text across files in the workspace.
- **Source Control Icon (Branching):** Opens the Source Control panel for Git operations. If no folder is open or no repository initialized, it provides options like "Open Folder" or "Initialize Repository".
- **Run and Debug Icon (Bug with Play):** Opens the Run and Debug panel to configure and manage debugging sessions, set breakpoints, and control execution flow.

- **Extensions Icon (Blocks):** Opens the Extensions panel to browse, search, install, and manage VS Code extensions from the Marketplace. Features a "Search Extensions in Marketplace" input field.

**4. Integrated Terminal Panel**

- Accessed via View > Terminal or Terminal > New Terminal or Ctrl+`.
- Appears typically at the bottom of the window.
- Contains tabs: PROBLEMS, OUTPUT, DEBUG CONSOLE, TERMINAL, PORTS.
- The TERMINAL tab provides a command-line interface (shell).
  - Displays a prompt (e.g., `user@machine:~$`).
  - Allows typing and executing shell commands.
  - Has a dropdown menu to select the default shell (e.g., bash, sh, zsh).
  - Includes icons for creating a new terminal (+), splitting the terminal, and killing (deleting) the current terminal instance.

**5. Command Palette**

- Opened via Ctrl+Shift+P (or F1) or View > Command Palette.
- A text input field appears at the top of the editor area.
- Type to search for and execute various VS Code commands (e.g., `Preferences: Color theme`, `Change Language Mode`).

**6. Settings UI**

- Accessed via File > Preferences > Settings or Ctrl+,.
- Opens in a new tab.
- Features a search bar to find specific settings.
- Settings are grouped into categories (User, Workspace) and sections (Commonly Used, Text Editor, Files, etc.).
- Provides GUI controls (dropdowns, input fields, checkboxes) to modify settings.
- An icon (document with `{}`) in the top-right of the Settings tab allows opening the `settings.json` file for direct text-based configuration.

**7. Editor Area**

- Main area where file contents are displayed and edited.
- Supports multiple tabs for open files (e.g., "Untitled-1", `yourfile.txt`).
- New, empty files may show placeholder text like "Select a language...".
- The left margin (gutter) is used to set/unset breakpoints (typically by clicking).

**8. Dialogs & Pop-ups**

- **"Save As" / "Create File" Dialogs:** These dialogs (sometimes integrated into VS Code, sometimes OS-native) require typing a filename into an input field and then clicking a confirmation button (e.g., "Save", "Create File").

*Figure 10.* **Automatically generated usage manual during self exploration** on VScode.

**GIMP (GNU Image Manipulation Program) Usage Manual**

This document outlines the basic UI elements and functionalities of GIMP relevant to your training.

**I. GIMP Interface Overview**

**1. Main Menu Bar (Top of Application Window)**

- Contains text-based menus: `File`, `Edit`, `Select`, `View`, `Image`, `Layer`, `Colors`, `Tools`, `Filters`, `Windows`, `Help`.
- Clicking a menu name reveals a dropdown list of commands.

**Example commands:**

- `File → New..., Open..., Quit`
- `Edit → Undo, Preferences`
- `Layer → New Layer...`

**2. Toolbox (Typically on the Left Side)**

**Tool Icons Grid**

A collection of icons, each representing a specific tool. Clicking an icon selects that tool and makes it active.

**Examples of Tools and Their Icons:**

- `Rectangle Select Tool`: Dashed rectangle (first icon)
- `Ellipse Select Tool`: Ellipse icon
- `Free Select (Lasso) Tool`: Lasso shape
- `Fuzzy Select (Magic Wand) Tool`: Magic wand with sparkles
- `Select by Color Tool`: Wand pointing at three RGB squares
- `Scissors Select Tool`: Scissors icon
- `Paths Tool`: Pen nib drawing a curve
- `Color Picker Tool`: Eyedropper icon
- `Zoom Tool`: Magnifying glass
- `Move Tool`: Four-way arrows
- `Align Tool`: Central square with arrows
- `Crop Tool`: Scalpel or intersecting L-shapes
- `Text Tool`: Capital "A"
- `Bucket Fill Tool`: Paint bucket
- `Pencil Tool`: Pencil

- `Paintbrush Tool`: Paintbrush
- `Eraser Tool`: Pink eraser
- `Airbrush Tool`: Spray can
- `Ink Tool`: Fountain pen nib
- `Clone Tool`: Rubber stamp
- `Heal Tool`: Crossed band-aids

**Color Area (Below Tool Icons)**

- **Foreground Color Swatch:** Top color square. Click to open color picker.
- **Background Color Swatch:** Bottom square. Click to change background color.
- **Default Colors Icon:** Small overlapping black & white squares. Resets colors.
- **Swap Colors Icon:** Bent arrow icon. Swaps foreground and background.

**3. Tool Options Panel (Typically Below the Toolbox)**

- Shows settings for the currently selected tool.
- Dynamically updates based on tool selection.

**Common UI Elements:**

- Checkboxes
- Sliders
- Dropdown menus
- Input fields
- Buttons

**Example (Crop Tool):**

- 'Delete cropped pixels' (checkbox)
- 'Allow growing' (checkbox)
- 'Aspect ratio' (dropdown)

**Tool Options Buttons (usually at the bottom):**

- Save options
- Restore options
- Delete saved options
- Reset to default (often a tool + arrow icon)

**4. Image Area (Center)**

- Displays and allows editing of image files.
- This is the main canvas area.

**5. Dockable Dialogs (Typically on the Right Side)**

A collection of panels, organized with tabs:

**Common Tabs:**

- `Brushes`
- `Layers`
- `Channels`
- `Paths`
- `Undo History`
- `Tool Options` (if not docked below toolbox)

**Dialog Buttons:**

- Usually found at the bottom of each panel (e.g., new layer button = '+' on a page icon)

**II. Common Operations**

1. **Access Menus:** Click menu name → click command to execute.
2. **Select Tools:** Click tool icon in Toolbox → Tool Options update.
3. **Change Tool Options:** Use sliders, dropdowns, checkboxes in Tool Options Panel.
4. **Manage Colors:**
   - Click swatches to change foreground/background color.
   - Click default colors icon to reset to black/white.
   - Click swap icon to switch foreground/background.
5. **Switch Tabs in Right Panel:** Click tab title (e.g., `Layers`, `Brushes`).
6. **Use Dialog Buttons:** Click icon buttons within active dialog panel (e.g., new layer button).

**III. Important Distinctions**

- **Main Menu vs. Dialog Buttons:**
  - `Layer > New Layer...` (Menu bar command)
  - vs.
  - `New Layer` icon button in Layers dialog
- **'Colors' Menu vs. Color Swatches:**
  - `Colors` menu: For image color adjustments (e.g., brightness, hue).
  - Swatches in Toolbox: For selecting paint colors.

*Figure 11.* **Automatically generated usage manual during self exploration** on GIMP.

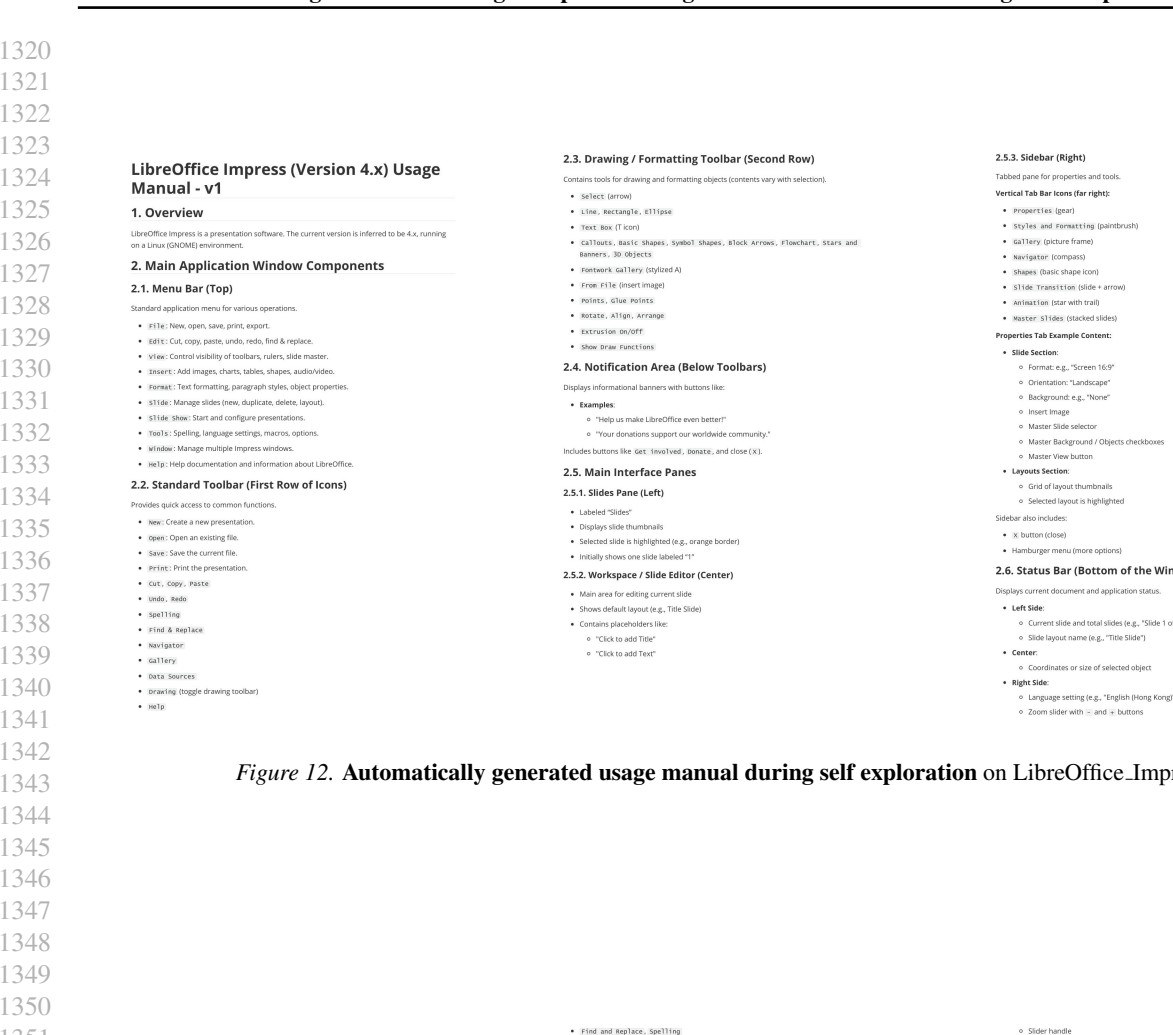

*Figure 12.* **Automatically generated usage manual during self exploration** on LibreOffice_Impress.

*Figure 13.* **Automatically generated usage manual during self exploration** on LibreOffice_Writer.