# OpenReview forum: "SEAgent: Self-Evolving Computer Use Agent with Autonomous Learning from Experience"
_ICML.cc/2026/Conference — ICML 2026 regular_

### Official Review · Reviewer_4wha · 2026-02-28

**Soundness:** 3
**Presentation:** 2
**Significance:** 3
**Originality:** 3
**Overall Recommendation:** 5
**Confidence:** 3

**Summary:**

This paper proposes SEAgent, a training framework designed to enable Computer Use Agents to achieve self-evolution on unfamiliar software through autonomous exploration. Its core innovation lies in constructing a closed-loop learning system comprising three key components: Actor Model, World State Model, and Curriculum Generator. It also introduces a specialist-to-generalist training strategy.

**Compliance With Llm Reviewing Policy:**

Affirmed.

**Final Justification:**

This paper addresses the problem of sparse reward signals in RL by proposing dense, step-level feedback provided by a specially trained "World State Model". Additionally, a curriculum generator is introduced to adaptively advance the exploration process. The motivation is sound and has been extensively evaluated. At first I had concerns about the generalization performance of the World State Model and the diversity of the curriculum generator. The authors addressed my concerns in the rebuttal stage, so I improved my score to "Accept".

**Key Questions For Authors:**

1. The paper mentions that the World State Model is fine-tuned using 860 trajectories annotated by GPT-4o. How do the authors believe this training enables the World State Model to generalize to entirely new environments?
2. When comparing with DigiRL and WebRL, could an ablation experiment be conducted where DigiRL/WebRL are also trained using the dense rewards provided by SEAgent's World State Model, to observe whether the RL method proposed in this paper yields significant improvements?

**Limitations:**

yes

**Strengths And Weaknesses:**

**Strengths:**

**S1. Reasonable motivation:** To address the issue of sparse reward signals in RL, the paper proposes dense, step-level feedback provided by a specially trained "World State Model." Additionally, a curriculum generator is introduced to adaptively advance the exploration process.

**S2. Comprehensive experiment:** Extensive evaluations are conducted on OSWorld, with comparisons against multiple strong baselines, and the method's cross-domain and cross-platform generalization is validated.

**Weaknesses:**

**W1. Methodological limitations and concerns:** The "World State Model" relies on training with specific data, and its judgments—especially in unseen environments—may contain errors, which could be amplified and further used to train the actor. Similarly, the task-evolving process is also generated by a model, making it difficult to ensure the quality and diversity of the tasks.

**W2. Limited evaluation of the curriculum generator:** Although the paper demonstrates the final performance of the curriculum generator compared to other task generation strategies (Table 5), it lacks an analysis of the quality and diversity of the generated tasks themselves.

**W3. Clarity or presentation issues:** The figures and tables are overly crowded; a more reasonable layout would further enhance the paper's readability. The expression of the paper could also be further simplified.

---

> ### Author Rebuttal · Authors · 2026-03-30
>
> We thank the reviewer for the careful reading and the positive assessment of our paper. We also appreciate the constructive questions on the World State Model and the curriculum generator. Below we respond to each point. All the discussion here will be included in the revision.
> ******
> ### Q1-W1-1. Evidence that the World State Model can generalize to entirely new environments
>
> We thank the reviewer for raising this question. The 860 trajectories used to train the World State Model are all collected from Chrome-based tasks in OSWorld, i.e., browser tasks involving web-page interactions. The WSM is therefore trained on a relatively narrow supervision source.
>
> Table 1 in the paper shows that the resulting WSM generalizes beyond browser tasks. After training on these Chrome trajectories, the WSM improves judgment accuracy not only on AgentRewardBench, but also on OSWorld Prof/Office settings, which include software such as VSCode and GIMP. These results indicate that the WSM is not limited to the browser setting used for annotation.
>
> We additionally evaluated the trained WSM on AndroidWorld [1]:
>
> | Setting | Precision | NPV |
> | --- | :---: | :---: |
> | Before WSM training | 41.9 | 79.0 |
> | After WSM training | 58.4 | 84.1 |
>
> The precision and NPV both improve, which is consistent with transfer to a different mobile environment.
> ******
> ### Q2. Whether DigiRL/WebRL can also benefit from SEAgent's dense reward
>
> We appreciate this suggestion. Comparing different RL algorithms under the same dense reward signal is useful for separating the effect of the reward model from that of the optimization method.
>
> We have now conducted this comparison, and the results are shown below.
>
> | Method | VSCode | GIMP | Impress |
> | --- | :---: | :---: | :---: |
> | DigiRL | 43.7 | 45.4 | 19.6 |
> | WebRL | 36.5 | 37.7 | 20.4 |
> | DigiRL + WSM dense reward | 44.3 | 47.7 | 20.4 |
> | WebRL + WSM dense reward | 39.4 | 39.5 | 21.3 |
> | GRPO + WSM dense reward | 41.7 | 41.5 | 20.9 |
> | AI + GRPO + WSM dense reward (SEAgent) | 46.1 | 50.0 | 21.3 |
>
> The results show that adding WSM dense reward improves DigiRL and WebRL on most settings. The full method remains strongest overall.
> ******
> ### W2. Limited evaluation of the curriculum generator
>
> We appreciate this observation. In the current version, Table 5 compares our curriculum task generator against several task generation strategies used in prior work and shows the downstream effect of our generator. However, this does not directly evaluate the quality and diversity of the generated tasks.
>
> To address this concern, we additionally measure task diversity using a combined embedding-space metric. Specifically, for each task generator, we encode the same number of generated task descriptions using OpenAI `text-embedding-3-small`, and compute: (1) the normalized embedding variance (`Var_norm`), which captures global semantic spread, and (2) the normalized average pairwise cosine distance (`Dist_norm`), which captures local semantic dissimilarity. We then define the final diversity score as:
>
> `Diversity Score = 0.5 * Var_norm + 0.5 * Dist_norm`
>
> The comparison is shown below:
>
> | Task Generator | Var_norm | Dist_norm | Diversity Score |
> | --- | :---: | :---: | :---: |
> | WebRL (Qwen2.5-72B) | 0.36 | 0.40 | 0.38 |
> | WebRL (Gemini-2.5-Pro-Thinking) | 0.61 | 0.71 | 0.66 |
> | NNetNav (Qwen2.5-72B) | 0.48 | 0.50 | 0.49 |
> | NNetNav (Gemini-2.5-Pro-Thinking) | 0.70 | 0.64 | 0.67 |
> | Ours (Qwen2.5-72B) | 0.65 | 0.67 | 0.66 |
> | Ours (Gemini-2.5-Pro-Thinking) | 0.78 | 0.80 | 0.79 |
>
> The table shows that our curriculum generator has higher diversity scores than WebRL and NNetNav under this metric. Using Gemini-2.5-Pro-Thinking further increases the score. In the current paper, we use Qwen2.5-72B to keep the full system based on open models. We agree that this analysis is still limited, and we will note this more clearly in the revision.
> ******
> ### W3. Clarity and presentation
>
> We thank the reviewer for pointing this out. We agree that several figures and tables in the current manuscript are crowded. In the revision, we will simplify the figures, reduce non-essential content, and make the presentation more concise.
> ******
> [1] AndroidWorld: A Dynamic Benchmarking Environment for Autonomous Agents

---

> > ### Author Rebuttal · Reviewer_4wha · 2026-04-01
> >
> > Thank you very much for your detailed response. I will consider adjusting my score accordingly.

---

### Official Review · Reviewer_DmPr · 2026-03-10

**Soundness:** 3
**Presentation:** 3
**Significance:** 3
**Originality:** 3
**Overall Recommendation:** 4
**Confidence:** 3

**Summary:**

This paper proposes SEAgent, a framework for training computer-use agents to operate unfamiliar software without human demonstrations. The system combines an actor policy, a World State Model that interprets GUI states and evaluates trajectories, and a Curriculum Generator that produces new tasks during training. A key component is the World State Model, which analyzes GUI screenshots and provides dense step-level rewards for agent actions. This enables reinforcement learning with richer feedback than sparse success signals. The framework also introduces a specialist-to-generalist training strategy and demonstrates improvements on several GUI benchmarks. Among the proposed components, the World State Model appears to be the most important contribution, as it enables dense step-level feedback that makes reinforcement learning feasible in this setting.

**Compliance With Llm Reviewing Policy:**

Affirmed.

**Final Justification:**

I remain somewhat skeptical about improvements on out-of-training tasks, but it improves the performance of relatively small models on specific tasks and is still a solid and meaningful contribution. I will keep my original score.

**Key Questions For Authors:**

- Have you evaluated the trained agents on environments or benchmarks outside OSWorld to test generalization?

- Have you analyzed the alignment between the World State Model reward and actual task success in the environment?

- Why should the framework be considered self-evolving, rather than curriculum-guided reinforcement learning?

- Can you provide ablations separating guidebook updates and curriculum task generation in the task generator?

**Limitations:**

yes

**Strengths And Weaknesses:**

### Strength

- The paper introduces a World State Model that provides dense step-level feedback from GUI trajectories, which helps address the sparse reward problem in training computer-use agents.
- The work presents a complete training pipeline combining exploration, reward modeling, curriculum generation, and reinforcement learning.
- Experiments on OSWorld and related environments show consistent empirical improvements over prior methods.

### Weakness

- The reward model appears to be trained using OSWorld trajectories. Since Table 2 also evaluates on OSWorld environments, this may limit the evidence for generalization. While Table 1 suggests the reward model works in other settings, it would be more convincing if the main results included tasks outside OSWorld.

- The paper does not analyze whether the reward produced by the World State Model aligns with true task success in the environment. It remains unclear when the reward correctly reflects task progress and when it does not.

- The term “self-evolving” appears somewhat overstated, as the improvements mainly come from dense rewards and curriculum-based tasks rather than autonomous discovery by the agent itself.

- The effect of the task generator is not fully disentangled. The method combines guidebook updates and curriculum task generation, but the paper does not isolate the contribution of each component.

minor
- The definition of Computer Use Agent (CUA) appears repeatedly throughout the paper.

---

> ### Author Rebuttal · Authors · 2026-03-30
>
> We thank the reviewer for the careful reading and the positive assessment of our paper. We also appreciate the constructive questions on generalization, reward alignment, the contribution of guidebook updates. Below we respond to each point. All the discussions here will be included in the revision.
> ******
> ### Q1. Generalization beyond OSWorld
>
> We thank the reviewer for raising this question. We evaluate the base model UI-TARS-7B-DPO and the SEAgent model, which is trained only on OSWorld in the paper, on two additional environments: ScienceBoard [1] and AndroidWorld [2].
>
> | Method | ScienceBoard [1] | AndroidWorld [2] |
> | --- | :---: | :---: |
> | Before Training (UI-TARS-7B-DPO) | 3.4 | 33.0 |
> | After Training (SEAgent) | 11.3 | 35.5 |
>
> These results show that, compared with UI-TARS-7B-DPO, SEAgent transfers to additional environments and does not only improve within the training benchmark. This table will be included in the revised paper.
> ******
> ### Q2. Alignment between the World State Model reward and true task success
>
> We thank the reviewer for this question. Table 1 in the paper is intended to analyze this issue. In that table, we use the verifier built from OSWorld human labels as the ground truth for true task success in the environment. This verifier is based on executable verification functions, which provide an accurate and reliable signal for whether the task state is correct. We then evaluate the judgment accuracy of the World State Model against this ground truth.
>
> We will clarify this point more explicitly in the revision and add case analysis. Based on our current observations, failure cases mainly occur on harder tasks with longer horizons, where the World State Model can occasionally hallucinate progress or failure signals for intermediate states.
> ******
> ### Q3. Why we describe the framework as self-evolving
>
> We appreciate this question. We use the term self-evolving to emphasize that the framework forms a closed loop with minimal human effort: the agent explores the environment, the collected trajectories are evaluated automatically, and the resulting experience is used to produce new training tasks and further improve the policy. The key point is not that the system evolves without any model-based components, but that it improves through its own interaction data rather than human demonstrations. We will clarify this wording in the revision.
>
> ******
> ### Q4. Ablation on guidebook updates
>
> We thank the reviewer for this suggestion. We conducted an ablation on the guidebook update mechanism:
>
> | Method | VSCode | GIMP | VLC |
> | --- | :---: | :---: | :---: |
> | w/o guidebook update | 44.3 | 46.2 | 13.4 |
> | w guidebook update | 46.1 | 50.0 | 31.8 |
>
> The improvement is modest on VSCode and GIMP, but much larger on VLC. We believe this difference is because the task generator Qwen2.5-72B is already relatively familiar with software like VSCode and GIMP, and can therefore generate new tasks more easily. In contrast, for more novel software such as VLC, the guidebook helps the system build a better understanding of the interface and available operations, which leads to more suitable generated tasks. This suggests that guidebook updates are especially helpful in less familiar environments. This table will be included in the revision.
> ******
> [1] ScienceBoard: Evaluating Multimodal Autonomous Agents in Realistic Scientific Workflows.
>
> [2] AndroidWorld: A Dynamic Benchmarking Environment for Autonomous Agents.

---

> > ### Author Rebuttal · Reviewer_DmPr · 2026-04-03
> >
> > Thank you very much for your detailed response. Your clarifications addressed my main concerns, though generalization beyond OSWorld could be clearer. I will keep my positive score.

---

> > > ### Author Response · Authors · 2026-04-04
> > >
> > > We sincerely thank you for the constructive follow-up feedback and for keeping the positive score.
> > >
> > > Regarding the point that generalization beyond OSWorld could be clearer, we would like to make the evaluation protocol explicit: the SEAgent model is trained only on OSWorld in the paper, without any further finetuning or adaptation on ScienceBoard or AndroidWorld. We then directly test this OSWorld-trained model on the two external environments. Compared with UI-TARS-7B-DPO, it improves from 3.4 to 11.3 on ScienceBoard and from 33.0 to 35.5 on AndroidWorld. We therefore consider this a direct indication that the learned improvements transfer beyond the training benchmark.
> > >
> > > We agree that this point should be stated more explicitly, and we will add these results and the corresponding clarification to the revised paper so that the cross-environment generalization evidence is presented more clearly.
> > >
> > > Thank you again for the helpful feedback.

---

### Official Review · Reviewer_Kz3G · 2026-03-12

**Soundness:** 2
**Presentation:** 2
**Significance:** 2
**Originality:** 2
**Overall Recommendation:** 3
**Confidence:** 5

**Summary:**

This paper proposes SEAgent, which self-evolves computer use agent with autonomous experience learning to explore novel and specialized software. The authors claim that SEAgent can explore new software, learn through iterative trial-and-error, and progressively tackle auto-generated tasks organized from simple to complex. To this end, they develop a curriculum generator to generate tasks and a world state model for step-wise trajectory assessment. They also we design a specialist-to-generalist training strategy to integrate individual experiential insights. Experiments on OSWorld, Androidworld and ScienceBoard demonstrate effectiveness.

**Compliance With Llm Reviewing Policy:**

Affirmed.

**Final Justification:**

The author's response have resolved some of my concerns, including a stronger annotation model and more refined ablation experiments. As a result, I have raised my score.

However, I still have doubts about the validity of GPT-4o as a distillation mode for building world state model, and the authors have not addressed my concerns. Therefore, I tend to weakly reject it.

**Key Questions For Authors:**

Please see weaknesses.

**Limitations:**

yes

**Strengths And Weaknesses:**

Strengths

1. The ideas of self-evolving and autonomous experience learning are important in the domain of compute use agent.

2.The method utilizes the agent's trial-and-error learning in unknown environments, effectively eliminating the reliance on manually labeled data.

Weaknesses

1. The world state model is built upon GPT-4o annotations, which lack interpretability and may introduce inherent model biases.

2. GPT-4o is a mediocre model and lacks important ablation comparisons to demonstrate the effectiveness of other, more state-of-the-art models on constructing world state model.

3. The paper's figures are terrible, containing too much fancy and irrelevant information.

4. The paper lacks important ablation studies; why did the authors only report ablation results in the OSWorld VScode scenario?

5. The description of related work is seriously inadequate; the authors have overlooked a large amount of CUA work, such as Mobile-Agent-v3, UItron, etc.

[1] Mobile-Agent-v3: Fundamental Agents for GUI Automation
[2] UItron: Foundational GUI Agent with Advanced Perception and Planning

---

> ### Author Rebuttal · Authors · 2026-03-31
>
> We thank the reviewer for the detailed comments and for recognizing the importance of autonomous experience learning for computer-use agents. We also appreciate the concerns on the construction of the World State Model, the choice of the annotator model, the scope of the ablation study, the presentation of the figures, and the related-work coverage. Below we respond to each point. We will incorporate these clarifications and all the discussions in the revision.
> ******
> ### Q1. Why the World State Model is built from GPT-4o annotations
>
> Our goal in using GPT-4o annotations is to improve the correctness of the judge model. In our preliminary study, we found that Qwen2.5-VL is weaker at judging task completion from many high-resolution screenshots, especially when the visual context is long and slightly beyond its effective training context range. In contrast, GPT-4o has stronger long-context multimodal ability for this type of judgment task. We therefore use GPT-4o to provide supervision for a simple distillation process into Qwen2.5-VL.
>
> If possible, we would prefer human annotation for this step. However, annotating this amount of trajectory data at sufficient quality is expensive. Instead, we use a limited amount of high-quality model annotation to train the WSM. The resulting model is trained only on a small-scale Chrome browser trajectory-judgement pairs, while the evaluation in the paper shows improved judgment performance on other domain. These results suggest that the learned judging ability transfers beyond the annotation source.
>
> We agree that any model-based annotation may introduce limitations. Our results do not suggest that the WSM is merely reproducing narrow annotation artifacts from a single environment or its teacher model.
> ******
> ### Q2. Comparison with a stronger annotator model
>
> We perform distillation using Gemini3-Pro as the teacher model. The results are shown below:
>
> | Model | AgentRewardBench Precision | AgentRewardBench NPV | OSWorld-Full Precision | OSWorld-Full NPV | Prof/Office Precision | Prof/Office NPV |
> | --- | :---: | :---: | :---: | :---: | :---: | :---: |
> | Gemini3-Pro | 83.1 | 91.5 | 79.9 | 90.2 | 77.3 | 83.5 |
> | World State Model (Gemini3-Pro distilled) | 72.1 | 89.8 | 74.2 | 89.4 | 71.5 | 82.3 |
> | World State Model (GPT4o distilled) | 71.6 | 91.2 | 73.9 | 90.5 | 69.3 | 82.0 |
>
> These results show that although Gemini3-Pro itself is strong, its capability is not transferred perfectly to Qwen2.5-VL through distillation (also 860 annotations). At the same time, the gap between Gemini3-Pro-distilled and GPT4o-distilled WSM is small, so the conclusion that a distilled WSM can generalize beyond the Chrome annotation source remains the same.
>
> More broadly, proprietary multimodal models with strong long-context ability generally have the same practical issue for our setting: API-based inference is too slow and expensive to support the full scale of experiments in this paper within a reasonable time budget. In addition, we intentionally want the core training system to remain based on open-source models, so that the resulting framework is easier for future research to reproduce, analyze, and extend. For these reasons, proprietary models are not used directly as the main judge model in our work.
> ******
> ### Q3. Figures and presentation
>
> We thank the reviewer for this feedback. Our intention was to present the full training loop and all major system components in one place, but we agree that the current presentation is too dense. In the revision, we will simplify the layout and improve readability.
> ******
> ### Q4. Scope of the ablation study
>
> We understand the reviewer's concern about the limited ablation scope. Because the full system requires substantial compute, we initially reported the ablation on a representative subset. During the rebuttal period, we were able to supplement it with additional results on two more software environments. The updated ablation is shown below:
>
> | Qwen2.5VL-72B | World State Model | SFT (BC) | GRPO | AI | VSCode | GIMP | VLC |
> | :---: | :---: | :---: | :---: | :---: | :---: | :---: | ---: |
> |  |  |  |  |  | 30.4 | 34.6 | 11.8 |
> | ✓ |  | ✓ |  |  | 26.1 | 32.3 | 13.7 |
> | ✓ |  |  | ✓ |  | 28.3 | 33.8 | 11.8 |
> |  | ✓ | ✓ |  |  | 34.8 | 37.7 | 15.7 |
> |  | ✓ | ✓ |  | ✓ | 39.1 | 38.5 | 19.6 |
> |  | ✓ |  | ✓ |  | 43.5 | 42.3 | 25.5 |
> |  | ✓ |  | ✓ | ✓ | 46.1 | 50.0 | 31.8 |
>
> These results will be added to the revised paper. They show the same trend on the added software environments: the World State Model, GRPO, and AI components each contribute to the final performance, and the full system performs best overall.
> ******
> ### Q5. Related work coverage
>
> We thank the reviewer for pointing this out. We agree that the current related-work discussion is incomplete. We will expand it to include additional recent CUA and GUI-agent works, including Mobile-Agent-v3 and UItron, and clarify the relation between these methods and our setting.

---

> > ### Author Rebuttal · Reviewer_Kz3G · 2026-04-03
> >
> > Thank you for the reply and clarifications. I now have a better understanding of the motivation behind this work.
> >
> > The response have resolved some of my concerns, including a stronger annotation model and more refined ablation experiments. As a result, I have raised my score. However, I have reservations about the results obtained using Gemini3Pro as the distillation model; its weaker performance compared to GPT-4o is difficult-to-explain.

---

> > > ### Author Response · Authors · 2026-04-03
> > >
> > > Thank you for the follow-up comment. We would first like to clarify that the Gemini3-Pro-distilled model is not weaker than the GPT-4o-distilled model in a general sense. This is not the pattern shown by the full results. Instead, the Gemini3-Pro-distilled WSM is consistently slightly stronger in precision, while the GPT-4o-distilled WSM is stronger in NPV. The correct interpretation is therefore not that Gemini3-Pro distillation underperforms, but that the two distilled models exhibit different trade-offs.
> > >
> > > For completeness, we add the full comparison below:
> > >
> > > | Model | AgentRewardBench Precision | AgentRewardBench NPV | OSWorld-Full Precision | OSWorld-Full NPV | Prof/Office Precision | Prof/Office NPV |
> > > | --- | ---: | ---: | ---: | ---: | ---: | ---: |
> > > | GPT-4o | 72.1 | 92.2 | 74.6 | 95.2 | 70.4 | 85.3 |
> > > | Gemini3-Pro | 83.1 | 91.5 | 79.9 | 90.2 | 77.3 | 83.5 |
> > > | World State Model (Gemini3-Pro distilled) | 72.1 | 89.8 | 74.2 | 89.4 | 71.5 | 82.3 |
> > > | World State Model (GPT4o distilled) | 71.6 | 91.2 | 73.9 | 90.5 | 69.3 | 82.0 |
> > >
> > > With the full table, the difference becomes more interpretable. As direct judges, GPT-4o and Gemini3-Pro already have different characteristics. Gemini3-Pro is extremely strong in precision, improving over GPT-4o by +11.0 on AgentRewardBench precision (72.1 -> 83.1), +5.3 on OSWorld-Full precision (74.6 -> 79.9), and +6.9 on Prof/Office precision (70.4 -> 77.3). In contrast, GPT-4o is stronger in NPV, especially on OSWorld-Full and Prof/Office.
> > >
> > > Our interpretation is that Gemini3-Pro is a stricter judge. Some trajectories that Gemini3-Pro predicts as failures can still pass the relatively permissive OSWorld verifier, which lowers NPV. By contrast, GPT-4o is a more permissive judge, which leads to higher NPV. The direct-model difference is therefore already not a simple stronger/weaker relation.
> > >
> > > The same pattern largely carries over after distillation. The Gemini3-Pro-distilled WSM still has slightly higher precision than the GPT4o-distilled WSM across all three evaluation settings: 72.1 vs. 71.6 on AgentRewardBench, 74.2 vs. 73.9 on OSWorld-Full, and 71.5 vs. 69.3 on Prof/Office. The margin is not large, but it is consistent. Our interpretation is that Qwen2.5-VL-7B is already close to its capacity limit on this judgment task, so a stronger teacher does not translate into a proportionally larger distilled gain. In other words, once the teacher is already sufficiently strong, the bottleneck becomes increasingly the student capacity rather than the teacher quality.
> > >
> > > At the same time, GPT-4o-distilled WSM retains stronger NPV, while Gemini3-Pro-distilled WSM retains slightly stronger precision. This difference is also relevant to the downstream training objective. If one uses GRPO alone without the AI component proposed in our work, precision is arguably the more important property, so Gemini3-Pro is theoretically a stronger teacher. However, in our full system, the AI component is particularly useful when the agent is still weak, because it helps the agent reflect on and learn from failures. In that regime, NPV becomes more important, and an overly strict judge can over-penalize weak agents by labeling too many recoverable trajectories as failures.
> > >
> > > More broadly, this suggests that GPT-4o and Gemini3-Pro are complementary rather than contradictory teacher models. In principle, the best setup for our training pipeline may be to use Gemini3-Pro (or its distilled WSM) to identify correct trajectories and GPT-4o (or its distilled WSM) to identify incorrect ones. We will add the full table and this discussion to the revised paper.
> > >
> > > Thank you again for the helpful follow-up comment and for prompting us to clarify this point more carefully.

---

### Official Review · Reviewer_PfMm · 2026-03-13

**Soundness:** 2
**Presentation:** 3
**Significance:** 3
**Originality:** 2
**Overall Recommendation:** 5
**Confidence:** 4

**Summary:**

This work provides a method for training CUA models on applications/environments without human-data. The training consists of many components that work together through multiple phases to improve the model's familiarity and performances in these applications: (1) the actor model or policy; (2) a World state model which is used for rewards; and (3) a curriculum generator which proposes tasks to solve. The model is trained with a step-wise variant of GRPO on just successfully judged steps along with an adversarial objective to push the model away from steps classified as failures. The work also demonstrates that initializing a model from specialist models trained on differnet environments before RL across all environments works better than directly performing RL on all environments.

**Compliance With Llm Reviewing Policy:**

Affirmed.

**Final Justification:**

Given the new experiments the authors provide comparing step-wise vs. trajectory-level optimization, most of my concerns have been addressed and I have increased my score accordingly.

**Key Questions For Authors:**

Please see weaknesses above.

**Limitations:**

Yes

**Strengths And Weaknesses:**

Strengths:
1. The work provides a method to train CUA agents to use applications without human guidance or demonstrations.
2. The trained models show significant improvements over the models before finetuning and other baselines.
3. The insight that the specialist-to-generalist method of training on multiple environments outperforms direct RL on all the environments is interesting.

Weakness:
1. The GRPO optimization is done step-wise instead of trajectory-wise in this work, which is non-standard for multi-turn agentic RL. The paper does not sufficiently motivate this choice. Intuitively doing things step-wise seems worse, as there might be multiple possible actions at a particular step that could lead to overall trajectory success, but the step-wise reward mechanism only accepts one such action.
2. The paper brands the method as self-evolving, but the curriculum generator uses much stronger and different model (Qwen2.5 72B) than the one being trained (UI-TARS-7B-DPO). So, it might be a bit misleading to state that the model self-improves/evolves.

---

> ### Author Rebuttal · Authors · 2026-03-31
>
> We thank the reviewer for the careful reading, the positive overall assessment, and the constructive comments on the training objective and the wording of self-evolving. We appreciate the recognition of the empirical improvements and the specialist-to-generalist training strategy. Below we respond to the two main concerns. All the discussions here will be included in the revision.
> ******
> ### Q1. Why we use step-wise GRPO instead of trajectory-wise optimization
>
> We thank the reviewer for this important question. We agree that step-wise optimization is less standard than trajectory-wise RL in multi-turn agent settings, and that the motivation should be explained more clearly in the paper.
>
> Our reason for using step-wise GRPO is that computer-use training in unfamiliar software is long-horizon and extremely sparse-reward, which makes trajectory-level credit assignment noisy and sample-inefficient. In contrast, the World State Model provides localized step-level progress signals. More importantly, under our step-wise GRPO design, the supervision for each step is no longer a binary 0/1 signal. Instead, it is a continuous value in [0, 1] computed from the action type and action parameters. This makes the reward denser and gives the policy a more informative learning signal at each step. We therefore optimize at the same granularity as the reward signal, which makes the training process more stable in this setting.
>
> We also agree that, at a given step, there may be more than one action that can still lead to eventual success. Our intent is not to assume that only one action is valid. The World State Model evaluates whether the observed state transition reflects task progress, rather than matching the action to a single reference trajectory. This means the step-wise signal does not explicitly enumerate all valid actions, but it is also not restricted to exact action matching. In this sense, the design trades some action diversity for lower-variance credit assignment, which we found important for long-horizon GUI interaction.
>
> We also agree that the ability to first make a mistake, then recognize it, and finally correct it is important in practical deployment. However, under the step-limit setting used in OSWorld evaluation, such trajectories often do not translate into higher benchmark scores, because the extra recovery steps consume the limited interaction budget. As a result, a denser step-wise reward is more aligned with the benchmark setting, while the recover-after-error behavior may be more valuable in real-world use than what is fully reflected by the OSWorld score.
>
> We will revise the paper to explain this design choice and its limitation more explicitly.
> ******
> ### Q2. Why we describe the framework as self-evolving
>
> We thank the reviewer for pointing this out. We agree that the term self-evolving can be misleading if it is interpreted as meaning that the target policy improves entirely on its own without assistance from any other model.
>
> Our intended meaning is that the full multi-agent system forms a closed loop and evolves without human effort, rather than that a single model improves entirely on its own. The agent interacts with the environment, its trajectories are used to construct learning signals and generate subsequent training tasks, and the policy is then improved from this accumulated experience. In this sense, the system is experience-driven and self-improving at the training-loop level.
>
> The stronger model in the curriculum generator is used only to propose training tasks; it does not provide expert demonstrations or action-level supervision for the target policy. In our early-stage experiments, single-model self-evolution with Qwen2.5-VL was much less effective, and we found that combining a specialized CUA agent such as UI-TARS with a stronger open-source LLM brought larger gains. However, we also tested single-model self-evolution on newer Qwen3-VL models and observed clear improvement:
>
> | Model | VSCode SR before | VSCode SR after self-evolve |
> | --- | ---: | ---: |
> | Qwen3-VL-8B | 43.7 | 49.5 |
> | Qwen2.5-VL-32B | 10.4 | 16.5 |
> | Qwen2.5-VL-7B | 6.1 | 9.6 |
>
> These new results suggest that single-model self-evolution can also be effective when the base model is sufficiently capable, although this is not the main setting studied in the current paper. We will update these results in the revised paper. At the same time, we agree that the current wording can overstate the claim, and we will revise the paper to make this distinction more explicit and avoid ambiguity.

---

> > ### Author Rebuttal · Reviewer_PfMm · 2026-04-03
> >
> > Thanks for the clarifications and additional details. Though, the argument for 1 would be much stronger with some experimental evidence. I will maintain my current positive score.

---

> > > ### Author Response · Authors · 2026-04-06
> > >
> > > To further clarify the value of step-wise GRPO, we additionally compare it with multi-turn RL under the experimental setting of ARPO [1]. Specifically, we evaluate three reward designs:
> > >
> > > 1. trajectory reward from WSM judge success/fail;
> > > 2. trajectory reward from WSM judge + critical wrong step mask;
> > > 3. step-wise reward.
> > >
> > > Here, the "critical wrong step mask" means that for failed trajectories, we do not penalize steps before the critical wrong step. This already brings a clear improvement over using only trajectory-level success/failure reward, and itself supports the value of the World State Model in providing finer-grained training signals.
> > >
> > > The results are shown below:
> > >
> > > | Setting | SR@5steps | SR@10steps | SR@15steps | SR@25steps | SR@50steps |
> > > | --- | :---: | :---: | :---: | :---: | :---: |
> > > | trajectory reward from WSM judge success/fail | 8.2 | 14.1 | 23.5 | 25.9 | 30.6 |
> > > | trajectory reward from WSM judge + critical wrong step mask | 10.6 | 12.9 | 25.9 | 35.3 | 40.0 |
> > > | step-wise reward | 11.8 | 21.2 | 29.4 | 31.8 | 34.1 |
> > >
> > > These results help clarify the trade-off. When the step budget is small or moderate, step-wise RL performs better, because denser step-level reward provides more informative credit assignment. In particular, at 15 steps, step-wise RL still clearly outperforms both trajectory-level variants (29.4 vs. 25.9 and 23.5), which supports our choice under the standard benchmark setting.
> > >
> > > At the same time, when the step budget becomes much larger, multi-turn RL can better exploit recovery behavior: the agent may first make a mistake, then recognize it, and finally recover within the longer interaction horizon. This is why the trajectory-level method with critical wrong step mask becomes stronger at 25 and 50 steps. We view this as consistent with the reviewer's intuition that trajectory-level RL can better capture recovery from mistakes when the evaluation setting is sufficiently fault-tolerant.
> > >
> > > Overall, these results support both points at once: (1) under the standard step-limited benchmark setting, step-wise RL is more effective because it gives denser and more stable supervision; and (2) when the environment allows more room for recovery, multi-turn RL can better leverage error correction. We will add this experiment and discussion to the revised paper to clarify the motivation and trade-off of step-wise optimization.
> > >
> > > We sincerely thank the reviewer for the constructive follow-up feedback after our rebuttal. We will incorporate these additional results and clarifications into the revised paper.
> > >
> > >
> > > [1] ARPO: End-to-End Policy Optimization for GUI Agents with Experience Replay

---

### Decision · Program_Chairs · 2026-04-30

**Decision:**

Accept (regular)

**Comment:**

This paper was positively received by the reviewers, who highlighted several key strengths:

- It tackles an important problem—training computer-use agents without human demonstrations—through a well-motivated autonomous learning framework.
- The World State Model, providing dense, step-level rewards, is a compelling contribution that effectively addresses sparse feedback in RL.
- The overall pipeline, including curriculum generation and specialist-to-generalist training, is coherent and empirically validated with consistent improvements across benchmarks.

Reviewers also noted areas for improvement. Some design choices, such as reliance on external models and step-wise optimization, could be better justified, and additional ablations and generalization analysis would further strengthen the claims.

Overall, the paper presents a solid and impactful contribution. Therefore, I recommend acceptance.